# Themis controls T cell activation, effector functions, and metabolism of peripheral CD8+ T cells

Namrata Gautam[1,2], Lukasz Wojciech[1,2], Jiawei Yap[1,2], Yen Leong Chua[1,2], Eyan MW Ding[2], Don CN Sim[2], Alrina SM Tan[3], Patricia J Ahl[2,4], Mukul Prasad[1,2], Desmond WH Tung[1,2], John E Connolly[2,4,5], Giulia Adriani[3,6] ©, Joanna Brzostek[1,2], Nicholas RJ Gascoigne[1,2,7] ©

Themis is important in regulating positive selection of thymocytes during T cell development, but its role in peripheral T cells is less understood. Here, we investigated T cell activation and its sequelae using a tamoxifen-mediated, acute Themis deletion mouse model. We find that proliferation, effector functions including anti-tumor killing, and up-regulation of energy metabolism are severely compromised. This study reveals the phenomenon of peripheral adaptation to loss of Themis, by demonstrating direct TCR-induced defects after acute deletion of Themis that were not evident in peripheral T cells chronically deprived of Themis in dLck-Cre deletion model. Peripheral adaptation to long-term loss was compared using chronic versus acute tamoxifen-mediated deletion and with the (chronic) dLck-Cre deletion model. We found that upon chronic tamoxifen-mediated Themis deletion, there was modulation in the gene expression profile for both TCR and cytokine signaling pathways. This profile overlapped with (chronic) dLck-Cre deletion model. Hence, we found that peripheral adaptation induced changes to both TCR and cytokine signaling modules. Our data highlight the importance of Themis in the activation of CD8+ T cells.

## Introduction

Themis (or Themis1), is a 72.8-kD protein important in thymocyte development (Fu et al, 2009; Johnson et al, 2009; Kakugawa et al, 2009; Lesourne et al, 2009; Patrick et al, 2009; Ge et al, 2018). It is expressed exclusively in the T cell lineage, with maximum expression in pre-selection double-positive (DP) thymocytes. Expression decreases in post-selection single-positive (SP) thymocytes and this level is maintained in mature peripheral T cells

(Fu et al, 2009; Johnson et al, 2009; Kakugawa et al, 2009; Lesourne et al, 2009; Patrick et al, 2009). Its participation during early TCR signaling is indicated by its tyrosine phosphorylation within 30 s post TCR stimulation (Fu et al, 2009; Brockmeyer et al, 2011). Themis interacts with canonical TCR signaling regulators, including constitutive interactions with Grb2 (Johnson et al, 2009; Lesourne et al, 2009; Patrick et al, 2009; Brockmeyer et al, 2011) and Shp1 (Fu et al, 2013; Paster et al, 2015; Zvezdova et al, 2016), whereas after TCR engagement, it interacts with LAT via Grb2 (Brockmeyer et al, 2011; Paster et al, 2013). In addition to this, GWAS studies have shown genetic polymorphisms closely linked to Themis that are associated with several diseases (Dubois et al, 2010; Kim et al, 2015b; Davies et al, 2016; Inshaw et al, 2018).

Germline Themis-deficient mice have thymic selection defects, wherein the development of preselection CD4+ CD8+ double-positive (DP) thymocytes into post-selection single-positive (SP: CD4+, CD8− or CD4−, CD8+) thymocytes and their further maturation is blocked (Fu et al, 2009; Johnson et al, 2009; Kakugawa et al, 2009; Lesourne et al, 2009; Patrick et al, 2009). Two models of Themis function have been proposed. One classifies Themis as a negative regulator of TCR signaling, where it positively regulates Shp1 phosphatase, dampening TCR signaling in DP thymocytes in response to low-affinity peptides (Fu et al, 2013; Paster et al, 2015). This curtailment of the TCR signal allows DP thymocytes to escape from activation-induced apoptosis (negative selection), therefore facilitating positive selection (Fu et al, 2013; Fu et al, 2014). Another model proposes that Themis is a positive regulator of TCR signaling, negatively regulating Shp1 phosphatase; thus enhancing TCR signaling in DP thymocytes in response to low-affinity peptides. This enhancement of the TCR signal allows DP thymocytes to escape from death-by-neglect, therefore allowing them to develop into SP thymocytes, hence facilitating positive selection (Zvezdova et al, 2016; Choi et al, 2017a, 2017b).

[1]Translational Immunology Programme, Yong Loo Lin School of Medicine, National University of Singapore, Singapore, Singapore   [2]Department of Microbiology and Immunology, Yong Loo Lin School of Medicine, National University of Singapore, Singapore, Singapore   [3]Singapore Immunology Network (SIgN), Agency for Science, Technology and Research (A*STAR), Singapore, Singapore   [4]Institute of Molecular and Cell Biology (IMCB), Agency for Science, Technology and Research (A*STAR), Singapore, Singapore   [5]Parker Institute for Cancer Immunotherapy, San Francisco, CA, USA   [6]Department of Biomedical Engineering, Faculty of Engineering, National University of Singapore, Singapore, Singapore   [7]Translational Cancer Programme, Yong Loo Lin School of Medicine, National University of Singapore, Singapore, Singapore

Correspondence: micnrjg@nus.edu.sg
Namrata Gautam's present address is H Lee Moffitt Cancer Center and Research Institute, Tampa, FL, USA

In Themis germline KO mice, peripheral CD8[+] and CD4[+] T cell numbers are drastically reduced and there is an increased percentage of memory-phenotype T cells (CD62L[lo]CD44[hi]) relative to naïve-phenotype T cells (CD62L[hi]CD44[lo]), which can be attributed to lymphopenia-induced expansion (Lesourne et al, 2012). The percentage of Foxp3[+] regulatory T cells (Treg) is increased (Fu et al, 2009; Johnson et al, 2009). Because altered phenotype and functions of peripheral T cells in the germline KO are likely the result of their defective thymic development (Brzostek et al, 2020), a better model to study the role of Themis in peripheral T cells is a conditional KO (cKO) active after thymic selection. We therefore made a cKO using distal Lck (dLck) promoter–Cre to delete Themis, finding that it has an important role in cytokine signaling and in integrating these signals with TCR signals in CD8[+] T cells (Brzostek et al, 2020; Liu et al, 2022). Given the importance of Themis in TCR stimulation during thymic selection and maturation, it was a surprise that cKO CD8[+] T cell responses to antigenic pMHC in vitro were unimpaired. dLck-Cre is expressed after positive selection (Wildin et al, 1995), and we observed Themis deletion in cKO SP thymocytes (Brzostek et al, 2020). Therefore, peripheral T cells in the Themis dLck cKO (Themis cKO dLck) have been Themis-deficient since the late stages of thymic development, and could have become adapted to Themis deficiency. This could mask real effects on TCR signaling because of the absence of Themis. Moreover, peripheral Themis deficiency in CD4[+] T cells has disparate effects on activation of naïve as compared with differentiated Th1 CD4[+] T cells (Yang et al, 2022a). In addition, in response to an acute viral infection, Themis deficiency increased CD8[+] T cell differentiation into short-lived effector cells, but inhibited differentiation into memory cells (Tang et al, 2023). These differences could result from the different roles of Themis in regulation of subset-specific, TCR-dependent, and TCR-independent signals, but may also reflect adaptation to Themis deficiency during CD4[+] differentiation.

We developed a tamoxifen-mediated acute Themis deletion model (Cre-ERT2), enabling us to study the role of Themis without any confounding effect of thymic and peripheral adaptation. Analysis of mice with acute deletion of Themis revealed its role in regulating T cell activation, signaling, and effector functions in mature CD8[+] T cells. We also found that Themis influences the metabolism of T cells, one of the key factors modulating T cell effector function. To investigate the signal adaptation that had obscured Themis' role in TCR signaling in the chronic dLck-Cre model, we compared gene expression in the acutely (Cre-ERT2-mediated) Themis-deleted cells with chronically Themis-deleted cells from both dLck-Cre–mediated deletion and long-term tamoxifen-treated Cre-ERT2 Themis[fl/fl] mice. We found far more changes associated with chronic deletion, and these showed trends common between the two models of chronic deletion, indicating how adaptation to lack of Themis expression is mediated.

# Results

### Efficient in vivo deletion of Themis by tamoxifen

To elucidate the function of Themis in T cells deprived of this protein acutely, we bred OT-I TCR transgenic Themis[fl/fl] mice

(Brzostek et al, 2020) with Cre-ERT2 mice (Wu et al, 2011). Cre-ERT2 expresses a Cre recombinase–estrogen receptor fusion which is activated only upon binding to its synthetic analog, tamoxifen (Feil et al, 1997). OT-I TCR transgenic mice generate H-2K[b]-restricted, SIINFEKL peptide-specific CD8[+] T cells (Hogquist et al, 1994). The mice were bred with the Ai14 Cre activity reporter strain, harboring a knock-in where a STOP cassette flanked by loxP sites prevents transcription of the red fluorescent protein tdTomato (Madisen et al, 2010). Cre recombinase deletes the STOP codon, resulting in the expression of tdTomato, which serves as an indirect measure of Cre-mediated recombination. We generated OT-I[+/−] Themis[fl/fl] Ai14[+/−] Cre-ERT2[−/−] (henceforth OT-I Themis cWT) and OT-I[+/−] Themis[fl/fl] Ai14[+/−] Cre-ERT2[+/−] (henceforth OT-I Themis cKO TX) mouse strains. To induce deletion, 5 mg tamoxifen was administered for five consecutive days to both OT-I Themis cWT and OT-I Themis cKO TX mice (Fig 1A). Themis deletion was assessed 4 d after the last injection. We confirmed that tamoxifen treatment induced Cre-ERT2 activity and efficient Themis deletion (Fig 1B–D).

### Increased CD8 expression on OT-I Themis–deficient CD8[+] T cells

After Themis deletion, OT-I Themis cKO TX CD8[+] T cells showed comparable numbers and percentages with OT-I Themis cWT. Interestingly, they displayed increased surface expression of the CD8 co-receptor (Fig 1E), but expression of Vα2 (as a marker for OT-I TCR) was unchanged (Fig 1F). The percentages and numbers of naïve or memory CD8[+] T cells between Themis cWT and Themis cKO TX CD8[+] T cells were similar (Fig 1G and H). A similar phenotype was observed for splenocytes (Fig S1A–C). Acute Themis deletion significantly increased the amount of CD8 expressed on peripheral OT-I T cells, suggesting reduced responses to self pMHC (Park et al, 2007; Takada & Jameson, 2009; Brzostek et al, 2020).

### Reduced phosphorylation of Shp1 in Themis-deficient peripheral OT-I CD8[+] T cells

Themis is phosphorylated soon after TCR engagement (Fu et al, 2009; Brockmeyer et al, 2011). Therefore, we wanted to test if acute loss of Themis in peripheral T cells influenced early T cell activation events. We stimulated CD8[+] T cells from tamoxifen-treated OT-I Themis cWT and cKO TX with tetramers loaded with OVA-altered peptide ligands (APLs: variants of the OVA peptide with known abilities to bind and stimulate the OT-I TCR) (Hogquist et al, 1994; Alam et al, 1996; Daniels et al, 2006). OVA APLs used in this study are classified based on their TCR affinities: high-affinity peptide OVA, intermediate affinity peptides Q4 and Q4R7, and low-affinity peptides Q4H7 and G4. We found a slight decrease in ERK phosphorylation in the Themis-deficient cKO TX CD8[+] T cells upon stimulation with lower affinity peptides, suggesting reduced TCR signaling. Because Themis interacts with and regulates Shp1 (Fu et al, 2013; Paster et al, 2015; Zvezdova et al, 2016; Choi et al, 2017b; Mehta et al, 2018), we analyzed phospho-Shp1 (pY-564), finding a consistent decrease in Themis cKO TX CD8[+] T cells, even upon stimulation with different OVA APLs (Figs 2A and S2). However, we found statistically significantly reduced CD69[+] cells only upon

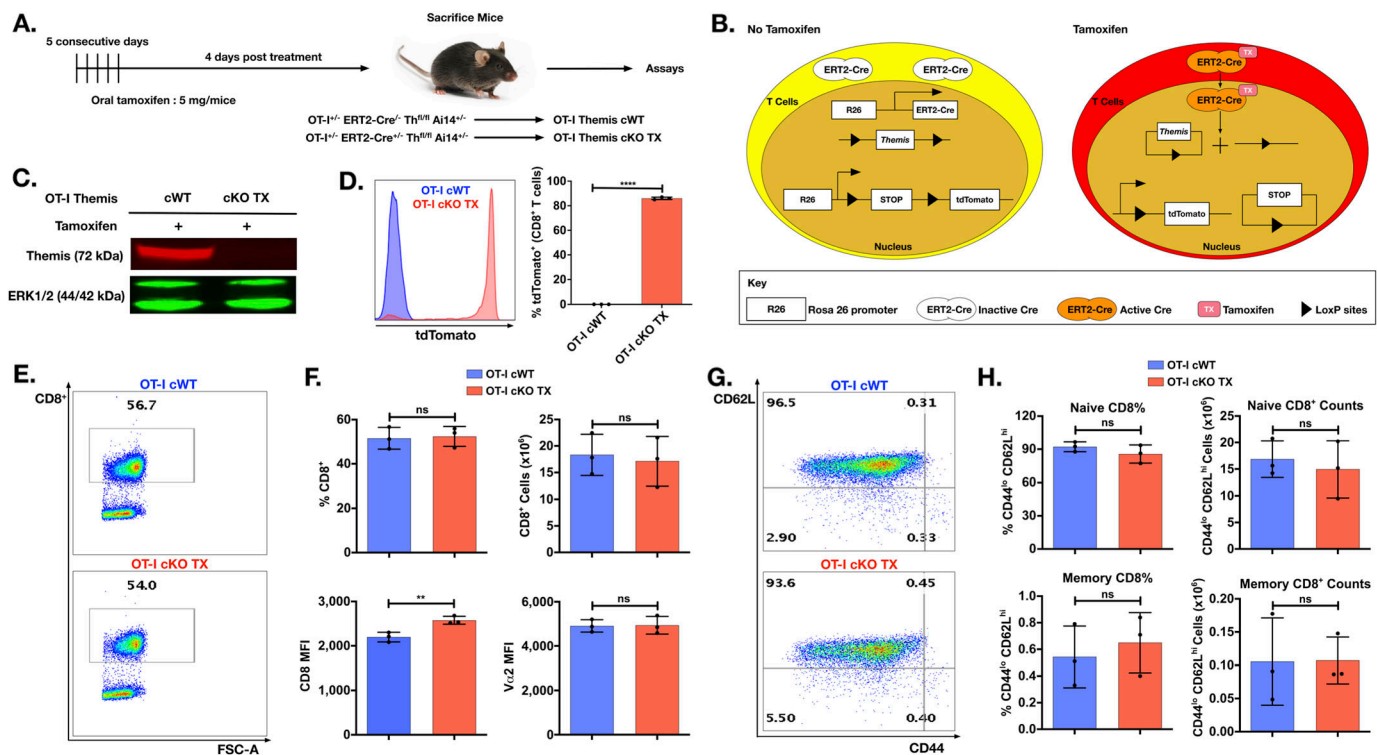

**Figure 1. In vivo Themis deletion using tamoxifen-induced Cre recombinase and phenotype of Themis-deficient CD8[+] cells.**
**(A)** Schematic representation of tamoxifen dose regimen followed to induce Themis deletion in peripheral T cells. **(B)** Schematic representation of mechanism of action of Cre ER–tamoxifen to induce simultaneous deletion of Themis and expression of tdTomato in OT-I CD8[+] T cells. **(C, D)** Expression of Themis and tdTomato, ERT2 activity reporter, in tamoxifen-treated OT-I Themis cWT and cKO TX. **(E, F, G, H)** Quantification by flow cytometry of CD8[+] T cells in the lymph nodes of B6 OT-I Themis cWT and cKO TX mice aged 7–12 wk upon tamoxifen treatment. **(E)** Dot plot showing the CD8[+] population in OT-I Themis cWT and cKO TX. **(F)** Average frequency, numbers, and CD8 expression and average surface expression of Vα2 of the CD8[+] population from the two groups. **(G)** Dot plot showing the naïve and memory of CD8[+] population in OT-I Themis cWT and cKO TX. **(H)** Average frequencies and numbers of naïve and memory subsets of CD8[+] T cells from two groups. Data from three mice per genotype. Data are representative of five experiments. OT-I Themis cWT and cKO TX samples were compared using t test, with **** indicating P-value < 0.0001 and ** indicating P-values from 0.001 to 0.01. Values shown here are mean ± SD.
Source data are available for this figure.

stimulation with the weak ligand Q4H7 (the weakest ligand tested, G4, showed little up-regulation of CD69) (Fig S3). Vα2 down-regulation between OT-I Themis cWT and cKO TX CD8[+] T cells upon stimulation with OVA APLs showed minor differences that switched between slightly stronger down-regulation in response to the higher affinity ligands to slightly less down-regulation in response to the lower affinity ligands. CD8 showed increased baseline expression in unstimulated Themis-deficient CD8[+] T cells, which remained higher in OVA APL-stimulated cells. We conclude that Themis influences Shp1 phosphorylation status, so that loss of Themis results in lower phosphorylation of Shp1 and ERK upon stimulation. However, this does not strongly alter CD8[+] T cell responses to short-term activation in vitro.

### Reduced proliferative capacity and IL-2 production in Themis-deficient peripheral OT-I CD8[+] T cells

Our results suggested that Themis-deficient OT-I CD8[+] T cells experience reduced activation, as indicated by the decreased phosphorylation of ERK1/2 and Shp1 upon stimulation with OVA APLs. We therefore asked if this would affect physiologically relevant outcomes of T cell activation, such as proliferation. We found no change in the percentage of divided cells upon stimulation with the high-affinity peptide OVA, but a subtle though significant decrease with the weaker Q4R7 peptide. There was a remarkable reduction upon stimulation with the low-affinity peptide, Q4H7 in OT-I Themis cKO TX CD8[+] T cells, as compared with the OT-I Themis cWT CD8[+] T cells (Fig 2B). This result clearly indicates that Themis regulates proliferation of activated T cells, with a greater influence on T cells interacting with the weaker affinity peptide ligands.

As proliferation was affected by Themis, we tested whether CD25 up-regulation (to form the high affinity IL-2R) and IL-2 production were hampered by Themis deficiency by stimulating the cells for 3 d with different OVA APLs, thus bringing about reduced proliferation in Themis-deficient CD8[+] T cells. CD25 expression was significantly lower in Themis cKO TX OT-I CD8[+] T cells compared with the control Themis-sufficient cells for the higher affinity ligands, and lower, though not statistically different, for the weaker ligand Q4H7. The cKO was significantly weaker in IL-2 production for all ligands tested (Fig 2C–E). Therefore, we conclude that Themis deficiency reduces proliferation, CD25 up-regulation, and IL-2 production upon TCR engagement.

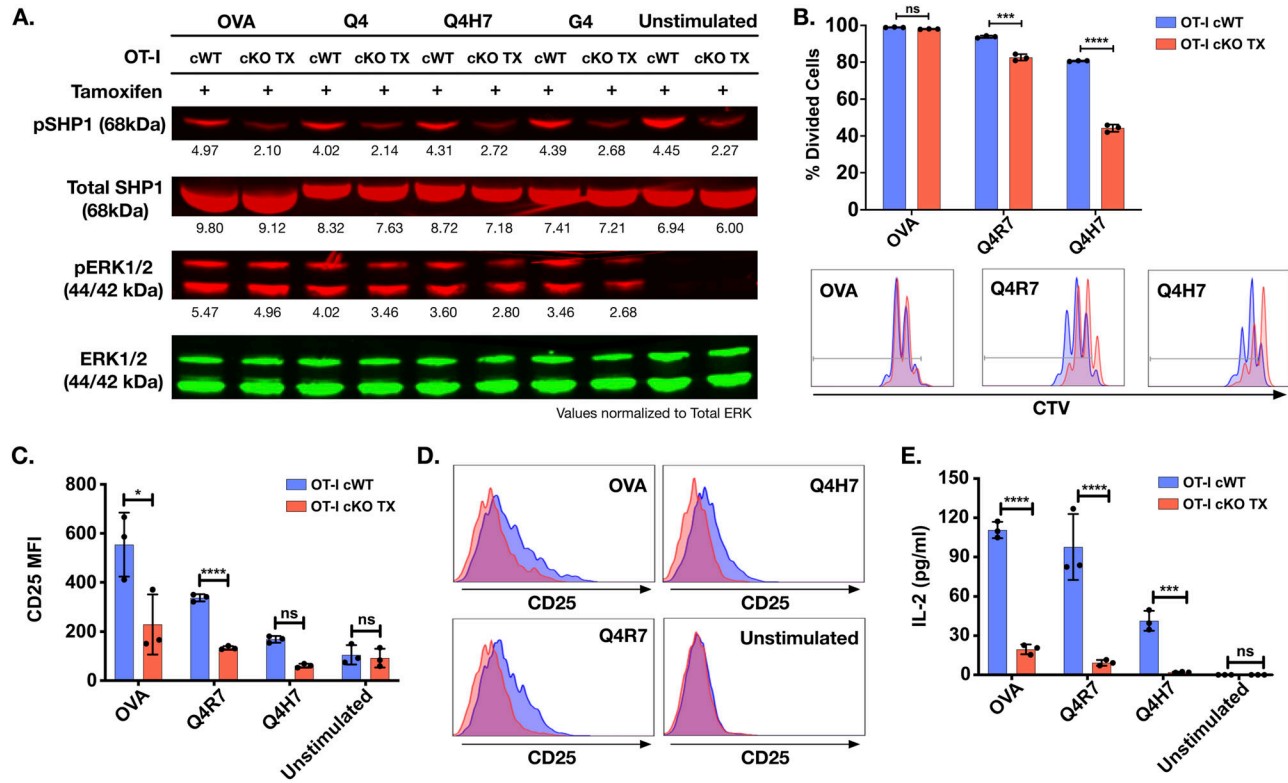

**Figure 2. Impaired activation, proliferative responses, and IL-2 production in OT-I CD8⁺ Themis-deficient T cells upon TCR engagement.**
**(A)** Western blot analysis of Shp1 and ERK1/2 phosphorylation in tamoxifen treated OT-I Themis cWT and cKO TX CTL upon OVA APL tetramer stimulation for 2 min. Total ERK was used as loading control. Cells were pooled from three mice per genotype. Data are representative of three experiments. **(B)** Themis cWT and cKO TX OT-I CD8⁺ T cells were CTV labeled and stimulated with OVA APLs for 3 d. After 3 d, proliferation was measured by analyzing CTV dilution. Percentage-divided cells and histograms showing CTV dilution of Themis cWT (blue) and cKO TX (red) OT-I CD8⁺ T cells are shown. **(C)** Quantification of CD25 expression after 3 d of stimulation. **(D)** Overlaid histograms of Themis cWT (blue) and cKO TX (red) CD8⁺ T cells showing CD25 expression. **(E)** IL-2 secretion by Themis cWT and cKO TX CD8⁺ T cells after 3 d of stimulation. Cells were pooled from three mice per genotype. Data are representative of three experiments. Themis cWT and cKO TX samples were compared using two-way ANOVA with multiple comparison test, with **** indicating $P$-value < 0.0001, *** $P$-values from 0.0001 to 0.001, ** $P$-values from 0.001 to 0.01, and * $P$-values from 0.01 to 0.05. Values shown here are mean ± SD.
Source data are available for this figure.

## Reduced effector function of Themis-deficient peripheral OT-I CD8⁺ T cells

CD8⁺ T cells, upon recognizing their antigen, become activated and differentiate into CTL. The effect of Themis deficiency on CTL effector functions has not yet been addressed. CTL were generated from OT-I Themis cWT and Themis cKO TX mice after in vivo tamoxifen deletion, then tested for their cytotoxic capabilities and cytokine production upon stimulation with EL4 cells pulsed with OVA APLs. Loss of Themis caused slight (but significant) reduction of CTL activity upon stimulation with high-affinity peptides like OVA and Q4R7, but nearly a twofold reduction with lower affinity peptides like Q4H7 and G4 (Fig 3A and B). Cells were also assessed for degranulation by staining for CD107a, a lysosome-associated membrane protein found on the cell surface following stimulation-induced exocytosis of cytolytic granules. Similarly, we found reduced expression of the degranulation marker, CD107a (Fig 3C and D). Cytokine production, in particular IFNγ, is one of the hallmarks of CD8⁺ effector function. We found a remarkable decrease in IFNγ production in response to stimulation with higher and lower affinity peptides in Themis-deficient cells (Fig 3E). A similar trend was

evident for TNF production (Fig 3F). In addition, we found reduced numbers of cells expressing both IFNγ and TNF in Themis-deficient cells (Fig 3G). The gating strategy for all the data is presented in Fig S4A–C. Therefore, we conclude that Themis is crucial for CTL to perform their effector functions.

## Reduced effector functions in in vitro tamoxifen-treated Themis-deficient CTL

The data in the previous section showed the importance of Themis in regulating CD8⁺ T cell activation and effector functions. We used an in vitro deletion approach to make sure there was no adaptation of cells in vivo, wherein 4-OH tamoxifen (4-OHT) was used to mediate acute deletion of Themis in CTL, achieving nearly complete Themis deletion (Fig S5A and B). TCR and CD8 showed no change in surface expression after tamoxifen treatment (Fig S5C). As with in vivo Cre-ERT2 deletion, we found decreased cytotoxic activity upon loss of Themis (Fig S6A). Again, the magnitude of reduction was greater for low-affinity ligands (Q4H7 and G4). The amount of CD107a on the CTL surface after stimulation also showed a slight decrease upon Themis deletion (Fig S6B). Similar results were also

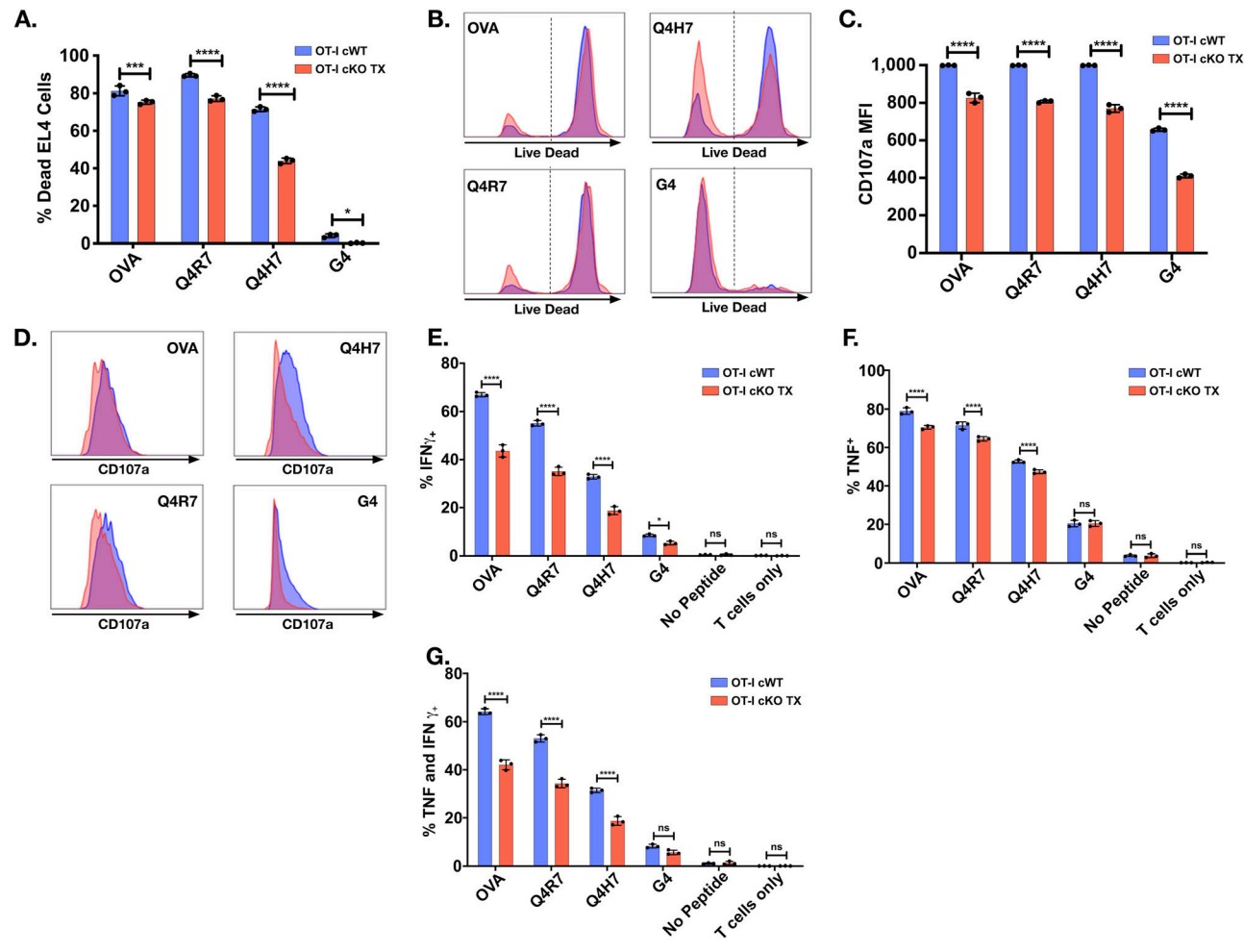

**Figure 3. Themis-deficient CTL have reduced cytolytic activity and cytokine production.**
In vitro cytotoxicity was assessed by co-culturing the OT-I CTL (effector) and OVA APL-pulsed EL4 (target) at a ratio of 10:1 (effector: target) for 4 h followed by live/dead staining and flow cytometry analysis. Cytokine production was induced by stimulating OT-I CTL (effector) with EL4 APCs (target) pulsed with OVA APLs at a ratio of 3:1 (effector:target) for 6 h. IFNγ and TNF expression was assessed via intracellular staining and analyzed by flow cytometry. Cells were also stained for degranulation marker, CD107a (after 6 h). **(A, B)** Specific lysis of antigen-pulsed EL4 by OT-I Themis cWT (blue) and Themis cKO TX (red) CTL represented using bar graph and histogram overlays. **(C, D)** CD107a expression on OT-I CTL from the same samples displayed using bar graphs and histograms. **(E, F, G)** Quantification of IFNγ, (F) TNF, and (G) double-positive OT-I CTL from Themis cWT and Themis cKO TX mice. CTL were pooled from three mice per genotype. Data are representative of three experiments. OT-I Themis cWT and cKO TX samples were compared using two-way ANOVA with multiple comparison test, with **** indicating *P*-value < 0.0001, *** *P*-values from 0.0001 to 0.001, ** *P*-values from 0.001 to 0.01, and * *P*-values from 0.01 to 0.05. Values shown here are mean ± SD.

found for reduced expression of IFNγ and TNF with ligands of varying affinities (Fig S6A–D). No physiologically significant changes or rather opposing trends were observed compared with the Cre-ERT2 group in the cytotoxic activity and cytokine production between the tamoxifen-treated and untreated controls (OT-I Themis cWT CTL, the right panel in each sub figure), therefore showing that the observed differences were because of Themis deficiency rather than tamoxifen treatment.

### Themis-deficient peripheral OT-I CD8$^+$ T cells show reduced energy-generating metabolism post activation

After activation of naïve T cells through TCR plus costimulation, many pathways emanating from these receptors drive T cells to differentiate, proliferate, and to perform their effector functions.

Recent advances in T cell biology show that metabolic reprogramming of T cells is critical for their effector functions. Coordination between T cell signaling events and metabolic programming directs the T cell response (Geltink et al, 2018; Balyan et al, 2020). We hypothesized that Themis could be involved in bringing about the metabolic changes required for activation of T cells, downstream of its role in TCR signaling. Upon activation, T cells need a boost of energy in the form of ATP to support increased biomass for growth and proliferation. CD8$^+$ T cells up-regulate the two main ATP-producing pathways, glycolysis and oxidative phosphorylation (OXPHOS), even before their first cell division (Geltink et al, 2018; Balyan et al, 2020). Naïve T cells fulfil their energy demands via OXPHOS, whereas upon activation, glycolysis is also initiated. Glycolysis is required for T cells to perform their effector functions (Prasad et al, 2021). In addition, mitochondrial ATP generated by

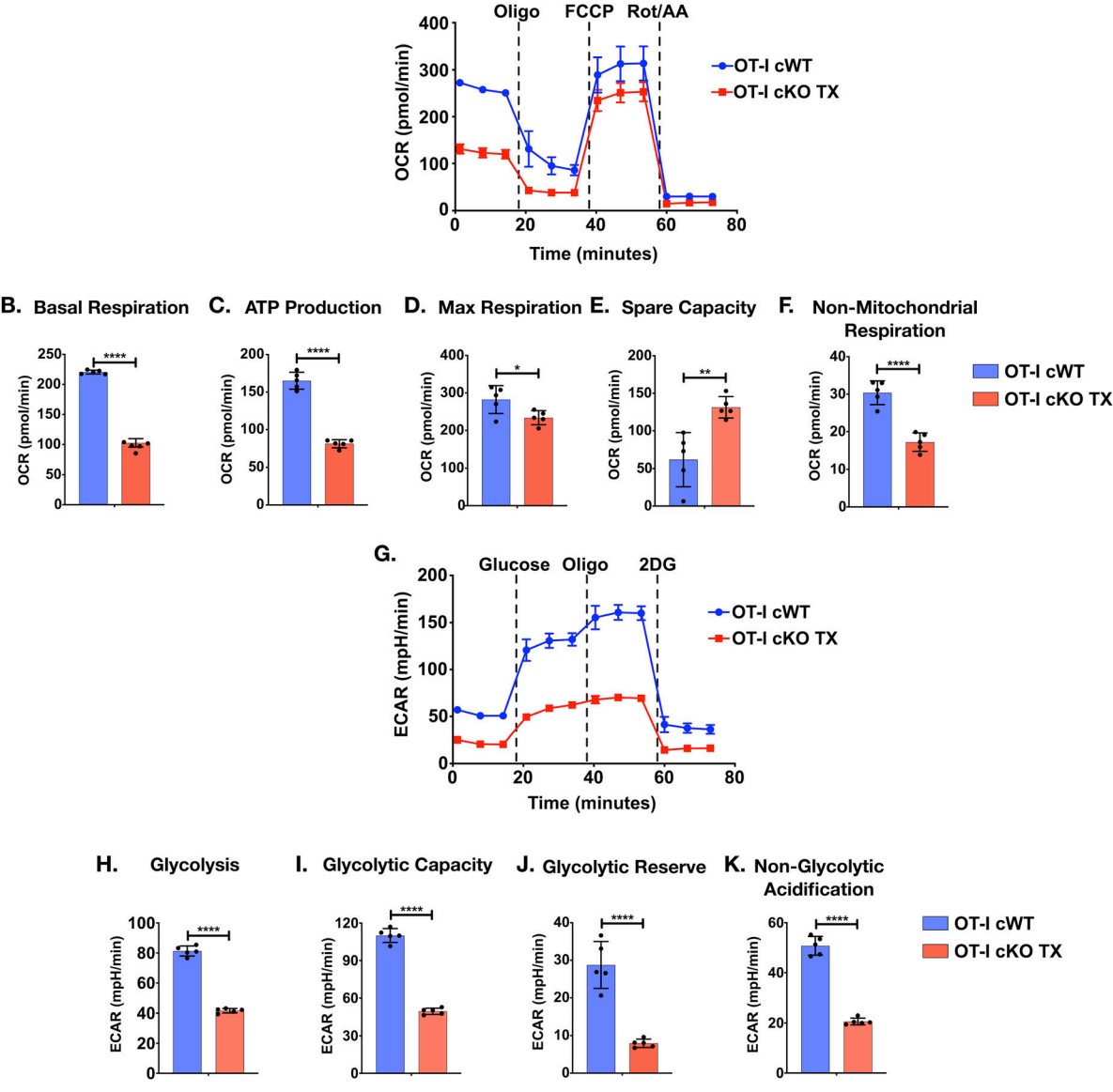

**Figure 4. Decreased energetics post activation in Themis-deficient CD8+ T cells.**
CD8+ T cells from OT-I Themis cWT and OT-I Themis cKO TX were stimulated with OVA for 24 h and mitochondrial and glycolytic parameters were quantified. **(A)** OCR of indicated cells at baseline and in response to oligomycin (Oligo), FCCP, and rotenone plus antimycin A (Rot/AA). **(B, C, D, E, F)** Quantification of basal respiration, ATP production, maximal (Max) respiration, spare capacity, and non-mitochondrial respiration in indicated cells. **(G)** ECAR of indicated cells at baseline and in response to glucose, oligomycin (Oligo), and 2-deoxy-D-glucose (2-DG). **(H, I, J, K)** Quantification of glycolysis, glycolytic capacity, glycolytic reserve, and non-glycolytic acidification in indicated cells. Cells were pooled from three mice per genotype. Data are representative of two experiments. Samples were compared using $t$ test, with **** indicating $P$-value < 0.0001, *** indicating $P$-values from 0.0001 to 0.001, ** indicating $P$-values from 0.001 to 0.01, and * indicating $P$-values from 0.01 to 0.05. Values shown here are mean ± SEM.

OXPHOS occurs 24–48 h after stimulation, associated with attainment of full activation and proliferative capacity by effector T cells (Geltink et al, 2018; Balyan et al, 2020). To test the role of Themis in energy generation, we stimulated OT-I Themis cWT and Themis cKO TX naïve T cells with OVA for 24 h and subjected them to mitochondrial or glycolytic stress using extracellular flux analysis.

The mitochondrial stress test showed that basal respiration in the stimulated Themis-deficient, cKO TX CD8+ T cells was less than half that of the Themis-sufficient, cWT CD8+ T cells (Fig 4A and B).

Therefore, absence of Themis lowered the energy use of the cells in general. Upon inhibition of the mitochondrial complex V ATP synthase with oligomycin, a twofold reduction was seen in oxygen consumption rate (OCR) of Themis-deficient compared with Themis-sufficient cells (Fig 4A and C). This suggested a decrease in the fraction of mitochondrial respiration contributing to ATP generation by Themis-deficient cells, as compared with the Themis-sufficient samples. Maximal respiration was measured by addition of carbonyl cyanide-4 (trifluoromethoxy) phenylhydrazone (FCCP),

which disrupts the proton gradient and collapses mitochondrial membrane potential. It was lower in Themis-deficient compared with Themis-sufficient cells (Fig 4A and D). Spare respiratory capacity, the difference between maximal respiration and basal respiration, showed a threefold increase in the Themis-deficient CD8+ T cells because, to begin with, basal respiration in these cells was lower; hence, it left more "room" to respond towards increased energy demands (Fig 4A and E). Mitochondrial respiration was abrogated by a mixture of rotenone, a complex I inhibitor, and antimycin A, a complex III inhibitor, allowing non-mitochondrial associated respiration to be measured. Lower non-mitochondrial respiration was seen in Themis-deficient compared with sufficient CD8+ T cells (Fig 4A and F). Therefore, we found that Themis-deficient CD8+ T cells showed decreased basal, mitochondrial, and non-mitochondrial respiration, but an increase in the spare respiratory capacity, in comparison with the Themis-sufficient cells.

To measure if Themis deficiency also affects T cell glycolysis, we performed a glycolysis stress test, measuring the extracellular acidification rate (ECAR). We found that Themis-deficient cells had twofold lower glycolysis as compared with the Themis-sufficient cells upon addition of glucose (Fig 4G and H). Oligomycin inhibits mitochondrial ATP production, compelling the cells to shut down OXPHOS and adopt glycolysis at maximum capacity, referred to as glycolytic capacity. Glycolytic capacity of cKO TX CD8+ T cells was barely above the amount before oligomycin was added, suggesting that the Themis-deficient cells could not up-regulate glycolysis as much as Themis-sufficient cells (Fig 4G and I). 2-deoxy-D-glucose (2-DG) blocks hexokinase and hence, glycolysis. Glycolytic reserve was calculated as the difference between the glycolytic capacity and glycolysis rate. Also, the ECAR levels before the glucose injection are reported as non-glycolytic acidification. A marked decrease was seen in the glycolytic reserve and non-glycolytic acidification in Themis cKO TX as compared with Themis-sufficient cells (Fig 4G, J, and K). Therefore, we found that Themis-deficient CD8+ T cells showed decreased glycolysis, glycolytic capacity, glycolytic reserve, and non-glycolytic acidification in comparison with the Themis-sufficient CD8+ T cells.

### Reduced metabolic gene expression in activated Themis-deficient CD8+ T cells

After T cell activation, TCR and metabolic signals lead T cells to enter a growth phase wherein T cells enhance their global metabolism, governed by phosphoinositide-3 kinase (PI3K), Akt, and mTOR signals (Powell et al, 2012; MacIver et al, 2013). mTOR induces c-Myc, which is essential for proliferation, cellular expansion, and effector functions (Wang et al, 2011; MacIver et al, 2013). mTORC1 also activates S6 kinase, which phosphorylates the S6 ribosomal protein required for efficient protein translation (Magnuson et al, 2012). Because we found that Themis-deficient CD8+ T cells have reduced proliferation, effector functions, and energy metabolism upon activation, we tested whether this reduction is because of Themis' role in activation of Akt/mTOR pathways.

We observed that upon stimulation of TCR, the amount of p-Akt and c-Myc were up-regulated in order of the pMHC affinities (Wu et al, 2023b), with OVA inducing the highest amounts and Q4H7 the lowest, for both Themis-sufficient and Themis-deficient OT-I CD8+

T cells (Fig 5A–F). The proportion of OT-I CD8+ T cells that were positive for p-Akt, c-Myc, and p-S6 upon TCR stimulation was reduced in Themis cKO TX mice compared with Themis cWT mice (Fig 5A, C, D, F, G, and H). Moreover, p-Akt and c-Myc-positive cells from OT-I cKO TX mice have lower MFI compared with Themis-sufficient cells from OT-I Themis cWT mice (Fig 5B, C, E, and F). Therefore, Themis modulates the amounts of molecular master regulators of T cell metabolism.

### Themis-deficient naïve OT-I CD8+ T cells have reduced mitochondrial mass after activation

Mitochondrial respiration is crucial for the generation of ATP upon T cell activation. Mitochondrial biogenesis is induced upon T cell activation, leading to increased mitochondrial mass, mainly mediated by PGC1α and c-Myc (Yuan et al, 2013). As we observed reduced OCR and c-Myc expression after cKO TX OT-I CD8+ T cell activation, we tested if Themis-deficient CD8+ T cells also show reduced mitochondrial biogenesis using MitoTracker Green (Gautam et al, 2018). After 72 h stimulation with OVA APLs, MitoTracker Green staining in OT-I Themis cKO TX CD8+ T cells was decreased as compared with the cWT cells (Fig 5I and J), indicating reduced mitochondrial mass. Hence, Themis contributes to mitochondrial biogenesis upon stimulation.

### Themis-deficient naïve OT-I CD8+ T cells show reduced expression of amino acid transporter

After TCR stimulation, mTORC1 and c-Myc promote amino acid uptake by enhancing expression of CD98, the large neutral amino acid transporter (Wang et al, 2011; Ren et al, 2017). We investigated if loss of Themis reduces CD98 surface expression in activated T cells. We stimulated naïve OT-I T cells with OVA APLs for 3 d. Themis-deficient CD8+ T cells showed reduced expression of CD98 upon stimulation as compared to Themis-sufficient T cells (Fig 5K and L).

Taken together, our data show that the metabolism of peripheral CD8+ T cells was severely compromised in the absence of Themis, with reduced up-regulation of each measure of metabolism that we have tested. ATP energy production was affected even without stimulation, where OXPHOS was reduced, and induction of glycolysis upon activation was strongly reduced. Biogenesis of mitochondria was reduced, as were the signaling cascades important in metabolism of activated cells and the ability to take up amino acids. These likely lead to reduced activation, proliferation, and effector functions. Moreover, dysregulated T cell activation and metabolic effects in CD4+ T cells have also been observed in a Themis germline KO model (Prasad et al, 2021). Therefore, Themis indeed plays a pivotal role in regulating T cell metabolism.

### Themis is required for anti-tumor activity of CTL

The importance of Themis in metabolism and effector functions led us to investigate if it also regulates the tumor-killing ability of CD8+ T cells. We designed a 3D T cell functional assay within a microfluidic tumor model (Pavesi et al, 2017) (Fig 6A) to test tumor killing and tumor-infiltrating capacities of CD8+ T cells. The microfluidic device consists of three parallel microchannels. B16 melanoma

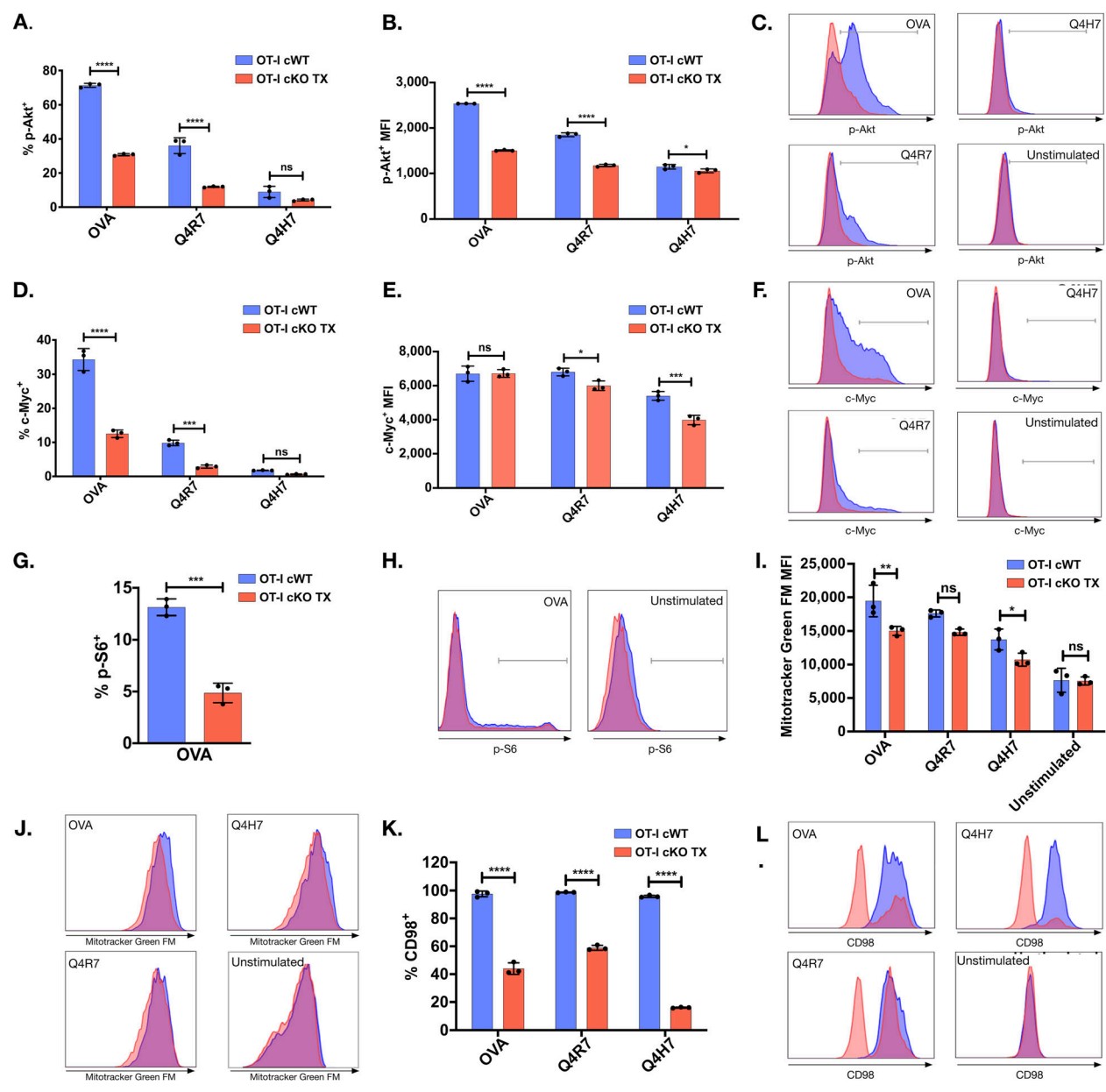

**Figure 5. Reduced metabolic regulators in activated Themis-deficient OT-I CD8⁺ T cells.**
**(A, B, C, D, E, F, G, H)** Naïve Themis cWT (blue) and Themis cKO TX (red) OT-I CD8⁺ T cells were stimulated with OVA APLs. After 24 h, p-Akt, c-Myc, and p-S6 were measured by flow cytometry in these groups. **(A)** Proportions of p-Akt-positive cells. **(B, C)** Bar graph summary of p-Akt MFI in p-Akt-positive cells and histogram showing the levels of p-Akt. **(D)** Proportions of p-Myc–positive cells. **(E, F)** Bar graph summary of p-Myc MFI in p-Myc–positive cells and histogram showing the levels of p-Myc. **(G)** Proportions of p-S6–positive cells. **(H)** Histogram showing levels of p-S6. Cells were pooled from 3 mice per genotype. (I-L) Naïve Themis cWT and Themis cKO TX OT-I CD8⁺ T cells were stimulated with OVA APLs for 3 d. **(I)** Quantification of mitochondrial mass by measuring MFI of MitoTracker Green FM by flow cytometry. **(J)** Overlaid histograms of Themis OT-I cWT (blue) and cKO TX (red) CD8⁺ T cells displaying amount of MitoTracker Green FM in the indicated cells. **(K)** Percentage of CD98 CD8⁺ T cells in Themis cWT and Themis cKO TX samples. **(L)** Overlaid histograms of Themis cWT (blue) and OT-I Themis cKO TX (red) CD8⁺ T cells showing CD98 expression. Cells were pooled from three mice per genotype. Data are representative of three experiments. Samples were compared using two-way ANOVA with multiple comparison test (all graphs except g) and *t* test (g), with **** indicating *P*-value < 0.0001, *** *P*-values from 0.0001 to 0.001, ** *P*-values from 0.001 to 0.01, and * *P*-values from 0.01 to 0.05. Values shown here are mean ± SD.

cells expressing either B16-OVA or B16-Q4R7 target cells were injected into the central gel channel and CellTrace Violet (CTV)-labeled OT-I Themis cWT or Themis cKO TX CTL were loaded into one of the lateral medium channels (Fig 6A and B). 24 h were allowed for the CTL to infiltrate into the tumor region and kill the

target B16 melanoma cells. Target cell-only controls were set up to monitor background cell death of tumor cells. After 24 h, dead fixable staining was performed, and samples were imaged. OT-I Themis cKO TX CTL showed significantly reduced cytotoxic response towards B16-OVA melanoma cells compared with the Themis-

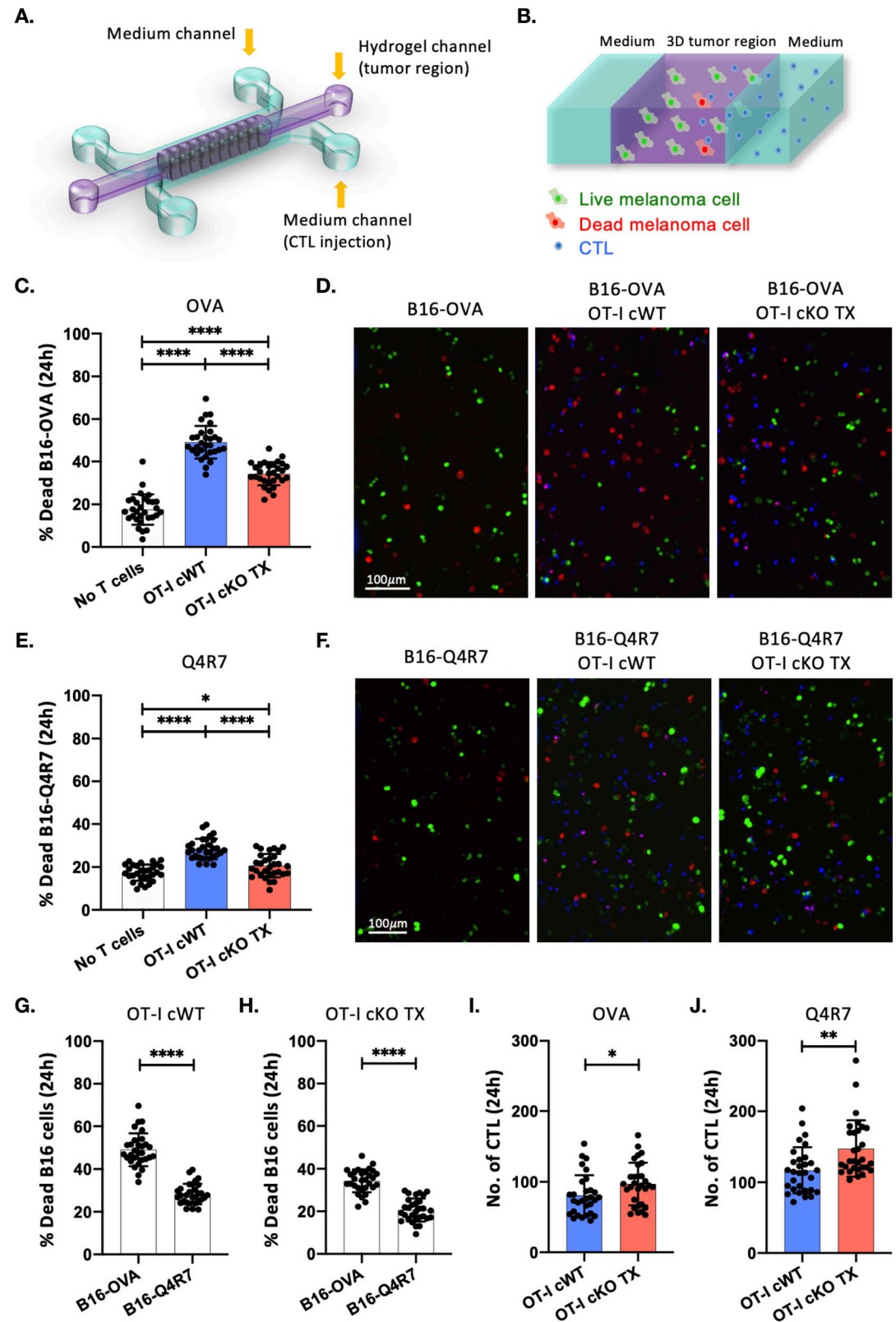

**Figure 6.   3D functional assay of CTL from Themis cWT and cKO TX OT-I mice.**
**(A)** Schematic layout of the microfluidic device to perform 3D co-culture of B16 melanoma cells and T cells from Themis cWT or cKO OT-I mice treated with tamoxifen (TX). The central channel (in purple) represents the tumor region, whereas the lateral channels (in cyan) provide nutrients and allow for CTL injection. **(B)** Zoomed-in schematic of the microfluidic system. Target melanoma cells were embedded in an extracellular matrix-like hydrogel and inserted in the central microchannel flanked by

sufficient control (Fig 6C and D), although OT-I Themis cKO CTL showed significantly higher infiltration (Fig 6I). Similarly, significantly reduced cytotoxic response was noticed towards B16-Q4R7 melanoma cells for OT-I Themis cKO TX CTL compared with the cWT CTL, but the infiltration was higher for the cKO cells (Fig 6E, F, and J). However, for both the groups cytotoxicity was higher upon interaction with B16-OVA as compared with the B16-Q4R7 (Fig 6G and H). Taken together these results support the hypothesis that Themis is important for tumor-killing function of CD8⁺ T cells as reflected by the consistent impairment in the cytotoxic ability of Themis-deficient CTL.

## Transcriptional adaptation to chronic Themis deficiency

Our finding that acute TX-mediated Themis deletion results in deficiencies in TCR signaling was in contrast to our previous results from a chronic dLck-Cre deletion model (Brzostek et al, 2020). We therefore performed mRNA sequencing to identify up-regulated and down-regulated genes in the different Themis-deletion scenarios. Differentially expressed genes (DEGs) were compared from CD8⁺ T cells from OT-I Themis cKO TX or cWT TX (Fig 7A upper panel: "acute deletion") and OT-I *Themis*^fl/fl^ dLck-Cre or the controls lacking the Cre module (Fig 7A lower panel: "chronic deletion"). A 10-fold larger number of DEGs was found because of the chronic (n = 1,300) rather than the acute (n = 127) deletion. Acute *Themis* deletion affected genes involved in a number of biological processes, notably including negative effects on cytokine signaling (JAK-STAT) and cell division, and up-regulation of genes involved in various stress responses (Fig 7B). We then analyzed the 1,300 genes identified as DEGs in the OT-I *Themis*^fl/fl^ dLck-Cre model for differential expression in acute TX-induced (Fig 7C upper left panel) or chronic TX-induced (Fig 7C upper right panel) *Themis* deletion. To achieve the chronic *Themis* deletion, the mice were rested for 4 wk instead of 4 d (as followed for the acute deletion model), post tamoxifen treatment before experimental use. There were many more genes common to the two chronic deletion models, with wider differences in expression, than between the Cre-ERT2 acute and dLck-Cre chronic models. Venn diagrams for the up- and down-regulated genes from these comparisons (Fig 7C lower panels), show a greater overlap between the two chronic models than between acute and chronic. Functional analysis of these DEGs revealed that changes induced in the two chronic deletion models are associated with increased regulation of lymphocyte or T cell activation (Fig 7D). Within the T cell activation-related DEGs, Themis (because of Cre-mediated deletion), Mitogen-activated protein

kinase binding protein 1 (*Mapkb1*), N-Myc and STAT interactor (*Nmi*), Casein kinase IIβ (*Csnk2b*), Calpain-1 catalytic subunit (*Capn1*), and Ikaros (*Ikzf1*), were down-regulated in both chronic Themis-deficient models (Fig 7E). The complete list of up- and down-regulated genes overlapping between the two chronic deletion models is shown in Table S1.

Mapkb1 enhances signaling by Jnk and reduces activation of NOD-2 (Lecat et al, 2012). Nmi is reported to increase STAT-mediated transcription in cytokine responses (Zhu et al, 1999). Casein kinase-2 is active in T cell signaling, through phosphorylation of CD5, affecting Erk signaling (Mier-Aguilar et al, 2016), and through the Akt and mTOR pathways (Yang et al, 2022b). It is also reportedly involved in positive regulation of SMAD-dependent pathways (Kim et al, 2018). Thus, down-regulation of this molecule might abrogate TGFβ-induced inhibition of T cell activation. Calpain-1 is a ubiquitous intracellular cysteine protease and is reported to be a negative regulator of NF-κB signal transduction, through its cleavage of IκB (Storr et al, 2011). Ikaros is a critical element of the T cell activation machinery by providing a threshold to entry into the cell cycle that is overcome by TCR signaling (Avitahl et al, 1999). Activation of CD8 T cells that express reduced amounts of Ikaros requires fewer TCR engagement events. In addition, these Ikaros-low cells are less sensitive to TCR and IL-2R signaling inhibitors (Avitahl et al, 1999).

The group of up-regulated DEGs in both chronic deletion models that are related to T cell signaling comprised only positive regulators of TCR-driven T cell activation (Fig 7E): the adaptor Grb2, the transcription factor Core binding factor *β* (*Cbfb*) (Wong et al, 2011), CD47 (Autio et al, 2022), and the chaperonin Hsp60, encoded by *Hspd1* (Albakova et al, 2022). Grb2 is a well-known adaptor molecule within the proximal TCR signalosome (Clements et al, 1999). It is constitutively bound to Themis (Johnson et al, 2009; Lesourne et al, 2009; Patrick et al, 2009; Brockmeyer et al, 2011; Paster et al, 2013). Core-binding factor *β* is involved with Runx1 in transcriptional control in T cell signaling, including expression of the TCR genes and T cell differentiation (Wong et al, 2011). CD47 acts through interaction with SIRPα to regulate TCR clustering during antigenic stimulation (Autio et al, 2022). Hsp60 is a mitochondrial chaperonin. Its expression is up-regulated after TCR signaling (Albakova et al, 2022).

Overall, our results suggest that immediately after Themis deletion, its loss causes reduced TCR signaling, presumably because of its absence from the TCR signalosome. Later, this negative effect of Themis deficiency on TCR signaling is compensated by controlled reprogramming of the expression of positive and negative

two lateral channels containing cell culture medium and CTL. Live and dead B16 melanoma cells are shown in green and red, respectively. CTL are shown in blue. **(C)** Scatter plot of the percentage of dead B16–OVA cells after 24 h from the injection of OT-I Themis cWT or cKO TX CTL in the fluidic microchannel. ****P < 0.0001. **(D)** Representative images of the 3D tumor region after 24 h from the injection of CTL (in blue) into the system. Live and dead B16–OVA cells are shown in green and red, respectively. **(E)** Scatter plot of the percentage of dead B16-Q4R7 cells after 24 h from the injection of OT-I Themis cWT or cKO TX CTL in the fluidic microchannel. *P = 0.0231, ****P < 0.0001. **(F)** Representative images of the 3D tumor region after 24 h from the injection of CTL (in blue) into the system. Live and dead B16-Q4R7 cells are shown in green and red, respectively. **(G)** Scatter plot of the percentage of dead B16-OVA and B16-Q4R7 cells after 24 h from the injection of OT-I Themis cWT CTL in the fluidic microchannel. ****P < 0.0001. **(H)** Scatter plot of the percentage of dead B16-OVA and B16-Q4R7 cells after 24 h from the injection of OT-I Themis cKO TX CTL in the fluidic microchannel. ****P < 0.0001. **(I)** Scatter plot of the number of OT-I Themis cWT and cKO TX CTL infiltrated after 24 h in the 3D tumor region with B16-OVA. *P = 0.0369. **(J)** Scatter plot of the number of OT-I Themis cWT and cKO TX CTL infiltrated after 24 h in the 3D tumor region with B16-Q4R7. **P = 0.0019. Mean ± SD values are shown. Each data point in the graphs represent the value for one region of interest. Data are from n = 5 regions of interest per device, n = 3 devices per condition and n = 2 independent experiments. Samples were compared by one-way ANOVA with multiple comparisons (C, E) and by unpaired two-tailed *t* test (G, H, I, J).

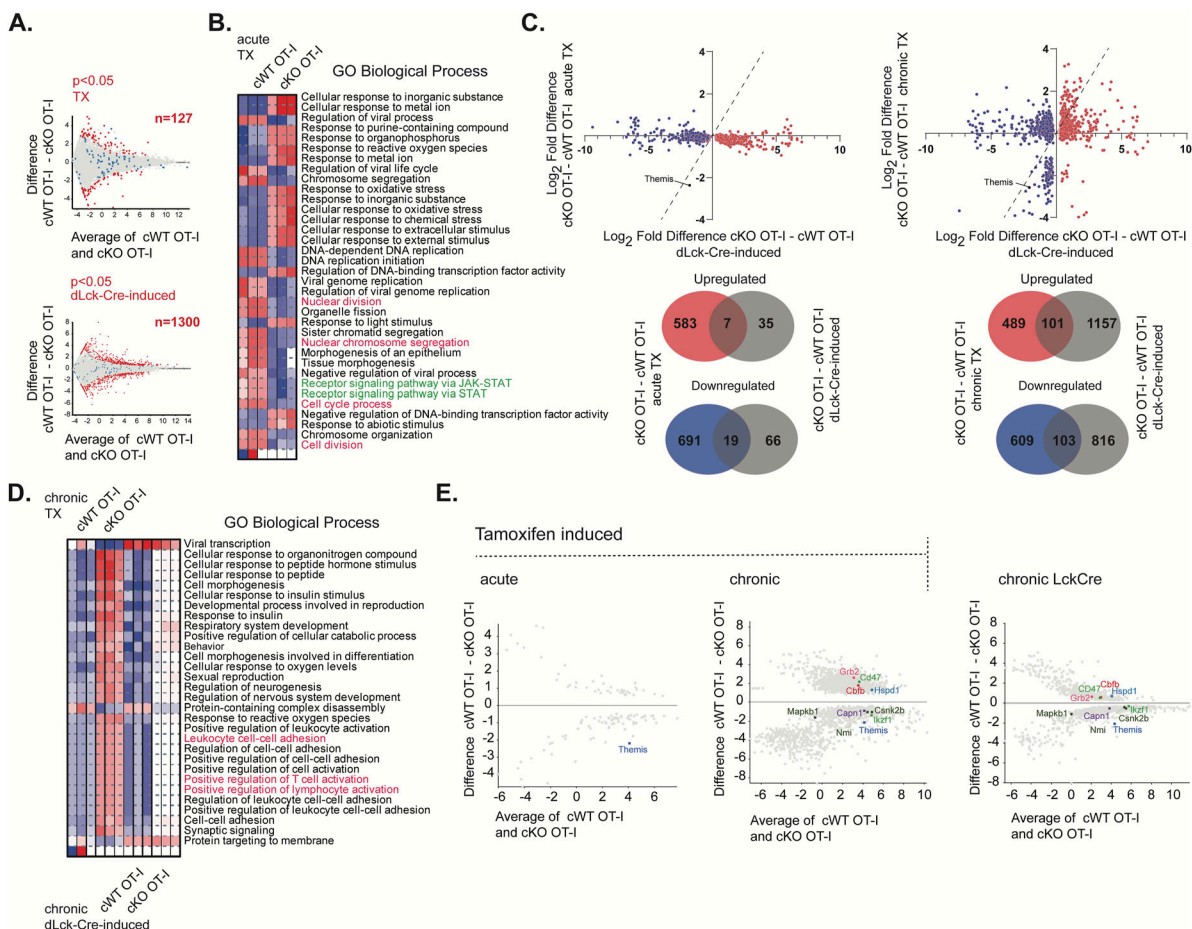

**Figure 7. Analysis of Themis-associated DEGs in mouse models in different Themis deletion models.**
**(A)** Analysis of Themis-associated DEGs in mouse models representing acute (*Upper panel*: cWT or cKO OT-I mice treated with tamoxifen [TX] after 4 d TX treatment) and chronic (*Lower panel*: dLck-Cre–induced Themis deficiency compared with Themis sufficient dLck-Cre-negative model) Themis deficiency scenarios. The DEGs between experimental groups were estimated using DESeq2 (Love et al, 2014) with *P* < 0.05, and the Themis-associated DEGs are highlighted in red. Blue dots represent DEGs between transcriptomes from Cre-negative animals that were treated or non-treated with TX. Three mice per group were used, and n indicates the number of DEGs identified. **(B)** Functional analysis of the transcriptomic changes induced in CD8[+] T cells by acute TX-driven Themis deletion in OT-I cKO. Themis-associated signatures from the experiment depicted in (A) (upper panel) were investigated for biological function using iDEP.96 platform (Ge et al, 2018) and GO Biological Process gene set as a reference. Blue indicates the negative effect of transcriptomic changes on a given process. **(C)** *Upper panel*: 3-dimensional comparison of DEGs between TX-induced acute Themis deficiency (left), TX-induced chronic (4 wk after start of TX treatment) Themis deficiency (right) to the DEGs from dLck-Cre model. The analysis was narrowed down into the Themis-associated 1,300 DEGs identified in the dLck-Cre model. *Lower panel*: Venn diagrams representing comparison of up-regulated and down-regulated genes from acute and chronic TX-induced Themis-deficient CD8[+] T cells in relation to Themis-associated signatures identified in dLck-Cre–induced Themis deficiency model. **(D)** Functional comparison of transcriptomic changes associated with chronic Themis deficiency induced by TX (in Cre-ERT2 model) and dLck-Cre. Analysis was done using iDEP.96 platform and GO Biological Process gene set. Blue and red indicate negative and positive effects, respectively. **(E)** MA plots representing DEGs in acute and chronic TX-induced models together with Themis-associated genes from chronic Lck-Cre group as TCR signaling regulators (according to the GO Biological Process gene set).

regulators involved in various aspects of TCR-dependent T cell activation.

# Discussion

In this study, we have used a Cre-ERT2 tamoxifen-inducible acute Themis deletion model to investigate the role of this protein in peripheral CD8[+] T cells. Our data indicate that Themis has a critical role in T cell activation, particularly in proliferation and effector functions. Moreover, our results indicate that Themis has a

substantial role in facilitating metabolic remodeling of T cells upon stimulation.

The effect of Themis deficiency was noticeable in some relatively early TCR signaling events because ERK phosphorylation was somewhat decreased. In contrast, CD69 up-regulation was not affected. Themis interacts with Shp1 and controls its activity. We found that Shp1 was less phosphorylated before and after activation in the absence of Themis. We found drastic reductions in proliferation, cytotoxicity, and cytokine production in T cells after Themis deletion. In some cases, for example, proliferation, this effect was reduced in responses to the highest affinity APLs in OT-I CD8[+] T cells. A recent study from our laboratory described defects in

CD4[+] T cell effector functions and metabolism in *Themis* germline KO mice (Prasad et al, 2021). Thus, findings from the *Themis* germline KO model could be substantially reflective of the peripheral compartment. In conclusion, these data highlight the importance of Themis in activation of CD8[+] T cells, particularly in response to low TCR affinity ligands.

T cell activation leads to a rewiring and boosting of cellular metabolism, to support cell growth, proliferation, and production of effector molecules (Geltink et al, 2018; Balyan et al, 2020). Apart from the direct involvement of Themis in TCR signaling, a previous study showed down-regulation of genes involved in vitamin, cholesterol, and carbohydrate metabolisms, and cell cycle, in Themis-deficient thymocytes (Johnson et al, 2009). Upon acute in vivo deletion of Themis in peripheral cells, we observed severe defects in T cell metabolism. Themis deficiency disrupted metabolic regulation starting after TCR signaling, reducing the amount of p-Akt, c-Myc, and p-S6 (mTORC1 activation). The disrupted c-Myc and mTORC1 signaling in turn led to reduction in expression of CD98, a transporter of large neutral amino acids (Wang et al, 2011; Ren et al, 2017). T cells deficient for the *Slc7a5*-encoded chain of CD98 are unable to activate mTORC1 signaling or to express c-Myc, indicating that these processes are dependent on CD98 (Ren et al, 2017). They do not respond well to TCR stimulation and do not undergo proliferative expansion, indicating that this amino acid transporter is essential for T cell–mediated immune responses. This may, in part, explain the defective proliferation and effector functions exhibited by the Themis-deficient CD8[+] T cells. This positive metabolic circuit is disrupted in the absence of Themis. The metabolic switch of activated T cells towards aerobic glycolysis from OXPHOS is essential for IFNγ production (Chang et al, 2013). However, to attain full activation and proliferative ability, T cells produce increased amounts of mitochondrial ATP via OXPHOS during the first 24–48 h post stimulation (Chang et al, 2013). As a result of this, T cells undergo mitochondrial biogenesis and up-regulate their mitochondrial mass after activation. Themis-deficient CD8[+] T cells showed reduced mitochondrial mass post stimulation. This could be linked to the decreased amount of c-Myc, which is one of the factors responsible for inducing mitochondrial biogenesis (Yuan et al, 2013). These defects led to decreased OXPHOS and aerobic glycolysis, causing a remarkable decline in the energy production required to achieve CD8[+] T cell proliferation and effector functions. Overall, we can conclude that dysregulation of principal molecular metabolic regulators, p-Akt, c-Myc, and p-S6 (indicating mTORC1 activation), after acute deletion of Themis, created a metabolic ripple effect severely affecting downstream targets like CD98 expression and mitochondrial biogenesis, eventually leading to an overall reduction in T cell energetics.

One of the major functions of CTL is to kill tumors and infected cells. To test CTL-killing activity we used a 3D functional assay to more closely recapitulate the mechanical and physical characteristics experienced by cells when they interact with an extracellular matrix and with other cells. The use of a microfluidic device provides further advantages, such as the exclusion of gravity-mediated cell interactions and a reduction of biological material (Sackmann et al, 2014). Recently, tumor-on-a-chip models have been designed to study interactions of cancer and immune cells in

a 3D matrix and to evaluate adoptive cell therapy, showing great promise for screening and developing cell-based therapeutic approaches against solid tumors (Adriani et al, 2016; Lee et al, 2018). Here, from the 3D co-culture of B16 melanoma cells and CTL from Themis cWT or Themis cKO OT-I mice treated with TX, we found that Themis-deficient CTL showed significantly reduced cytotoxic ability despite their significantly increased migration towards the target tumor cells. This could be related to Themis' (with Shp1) interaction with CD137 which reduces signaling through CD137-containing chimeric antigen receptors (CARs) (Sun et al, 2020), or to its effect on PD-1 activity through Shp1 or Shp2 (Kinosada et al, 2017; Tang et al, 2023).

The acute deletion model allowed us to identify critical functions for Themis in peripheral T cell activation and signaling, which were not found in the dLck-Cre Themis deletion model (leading to absence of Themis in T cells from the SP thymocyte stage onwards) (Brzostek et al, 2020). In the dLck-Cre-mediated cKO model in OT-I mice, we found negligible reduction in TCR stimulation with OVA APLs but altered memory versus effector cell differentiation in responses to acute viral infection (Brzostek et al, 2020; Tang et al, 2023). Because of the possibility that the T cells could change their sensitivity to stimulation (Grossman & Paul, 1992) over their lifespan after passing the SP thymocyte stage, this suggests that the cells had adapted their signaling functions to overcome a Themis defect.

The Themis regulation of Shp1 could provide an explanation for the effect of Themis deletion on TCR activation and metabolic events. The decreased Shp1 phosphorylation observed in Cre-ERT2–driven Themis-deficient peripheral CD8[+] T cells indicates that Themis-dependent regulation of Shp1 provides an underlying molecular mechanism behind the TCR activation and metabolic defects of acute Themis deletion. This is supported by our previous data from the dLck-Cre Themis deletion model, as reduced responses to self pMHC and cytokines in the Themis-deficient CD8[+] T cells were rescued by Themis plus Shp1 dLck-Cre–mediated double-KO, where, in the absence of Themis, Shp1 activation suppressed Akt activation (Brzostek et al, 2020). Therefore, it is likely that the same mechanism is working in the case of acute Themis deletion, resulting in severely defective proliferation, effector functions, and metabolic phenotypes upon acute Themis deletion. Hence, our data indicate that Themis acts as a negative regulator of Shp1 in peripheral T cells, as proposed by Love and colleagues for thymocytes (Zvezdova et al, 2016; Choi et al, 2017a, 2017b). However, it is becoming clear that Themis acts not only through Shp1 regulation of TCR signals, but also through effects on PD-1 (Kinosada et al, 2017; Tang et al, 2023) and CD137 signaling, and through cytokine responsiveness (Liu et al, 2022). These separate but integrated responses are not necessarily through a single biochemical mechanism.

Our data on transcriptional regulation in the two chronic Themis deletion models, dLck-Cre and long-term TX-induced, suggest that the initial stage of Themis deletion leads to reduction of TCR signaling because of the lack of Themis in the TCR signalosome. Later, the negative effect of Themis deficiency on TCR signaling is compensated by controlled reprogramming of the expression of positive and negative regulators involved in TCR-dependent T cell activation.

This study shows that Themis is important for optimum activation and performance of efficient effector functions in peripheral CD8+ T cells. Themis is particularly important in the sensitivity of T cells towards lower TCR affinity or avidity ligands. The differences in TCR sensitivity seen with the acute Cre-ERT2 deletion model but not the chronic dLck-Cre deletion model, lead us to propose a model whereby T cells in the dLck-Cre-mediated Themis deletion model had become adapted to the absence of Themis. Hence, only upon using the acute Themis deletion model, did we find clear defects in TCR stimulation.

Because Themis allows increased sensitivity to weaker ligands in the periphery, it could be useful to tweak or enhance the responsiveness of T cells towards weak affinity tumor-derived antigens. Differences in control of signaling pathways via normal TCR-CD3 versus CAR is important in improvement of CAR-T cells as therapeutics (Wu et al, 2020, 2023a). Indeed, signaling through CD137 (4-1BB)-containing CARs leads to reduced CAR phosphorylation because of recruitment of the Shp1–Themis complex (Sun et al, 2020). Therefore, Themis signaling could perhaps be used as a novel therapeutic immunomodulator to fine-tune T cell effector responses against tumors, infections or autoimmunity. Several studies indicate an association of certain single-nucleotide polymorphisms in the *Themis* locus with diseases. These include celiac disease, multiple sclerosis, and atopic dermatitis in humans (Dubois et al, 2010; Sawcer et al, 2011; Bondar et al, 2014; Kim et al, 2015b), whereas Themis-deficient mice are more susceptible to pulmonary tuberculosis while being protected from cerebral malaria (Torre et al, 2015). Thus, knowing the role of Themis in the periphery may provide some important leads to improve immune responsiveness towards particular diseases.

# Materials and Methods

### Mice

Experiments were performed on OT-I Themis cWT and OT-I Themis cKO TX or Themis cWT and Themis cKO TX (see text) 7–12-wk-old, sex, and age-matched mice.

OT-I *Themis*$^{fl/fl}$ Ai14$^{+/-}$ Cre-ERT2$^{-/-}$ and OT-I *Themis*$^{fl/fl}$ Ai14$^{+/-}$ Cre-ERT2$^{+/-}$ mice strains were generated using the following strains: *Themis*$^{fl/fl}$ and 129-Gt(ROSA)26Sortm1(cre/ERT2)Tyj/J, (strain #008463; The Jackson Laboratory). These mice were crossed with Ai14, Cre activity reporter strain (B6.CgGt [ROSA] 26Sortm14 [CAG-tdTomato]Hze/J, [strain #007914; The Jackson Laboratory]) and further bred to OT-I (C57BL/6-Tg[TcraTcrb]1100 Mjb/J, [strain #003831; The Jackson Laboratory]) to obtain OT-I Themis cKO TX and cWT.

These mice strains were bred and maintained under specific pathogen-free conditions in Comparative Medicine vivarium, National University of Singapore. All animal experiments were performed and approved according to the Institutional Animal Care and Use Committee at the National University of Singapore.

### In vivo tamoxifen treatment

To mediate acute TX-induced conditional deletion of *Themis*, mice between 7–12 wk of age were administered 5 daily doses of 5 mg tamoxifen emulsion via oral gavage, then rested for 4 d before experimental manipulation. For chronic deletion, the mice were rested for 4 wk after inducing tamoxifen-mediated *Themis* deletion.

### Metabolic analysis

Mitochondrial analysis was performed largely as described (Gautam et al, 2018). T cells were incubated with MitoTracker Green FM (Cat #M7514; Invitrogen), washed with PBS, and analysed by flow cytometry. OCR and ECAR were measured using a 96-well XFe Extracellular Flux Analyzer (Seahorse Bioscience) at $2 \times 10^5$ cells per well, using the manufacturer's protocols.

### Imaging

Z-stack images were acquired for five regions of interest for each microfluidic device, three devices for each condition. Image analysis to quantify number of live/dead target cells and infiltrated T cells was performed by IMARIS software (Bitplane).

### RNA sequencing

For the bulk RNA sequencing, total RNA was isolated using RNAeasy Kit (QIAGEN) from peripheral CD8+ T cells that were sorted based on CD8, CD122, CD62L expression. Synthesis of the libraries were performed using MagMAX DNA Multi-Sample Ultra Kit and sequencing of the obtained genetic material was done on an Illumina NovaSeq6000 instrument for sequencing using a 2 × 150 paired-end configuration according to the manufacturer's instructions. The raw reads were aligned to the reference genome using Hisat2 v2.1.0 (Kim et al, 2015a). SeqMonk v1.46.0 data analyzer was used to generate raw reads expression matrix. Further bioinformatics analysis was performed using RStudio software (posit.co/products/open-source/rstudio). Differential gene expression analysis was done using DESeq2 package with the significance cut-off FDR<0.05. Functional annotation of the DEGs was done using iDEP platform.

### Tamoxifen preparation

100 mg tamoxifen (T5648-5G; Sigma-Aldrich) was dissolved in 1 ml ethanol by incubating at 55°C for 10 min with intermittent vortexing after 5 min. Then, the tamoxifen was further diluted at 1:10 in corn oil (C8267; Sigma-Aldrich) to form an emulsion.

### Preparation of lymph nodes and spleens

Lymph nodes (cervical, axillary, branchial, and inguinal) and spleens were harvested from the dissected mice. To obtain a cell suspension, they were then mashed through a 70-$\mu$m strainer with 1 ml syringe plunger into 5 ml cRPMI RPMI medium, (Cat# SH30027.01; HyClone) supplemented with 10% (vol/vol) FCS (Cat #SV30160.03; HyClone), 100 U/ml penicillin, 10 mg/ml streptomycin (Cat #SV30010; HyClone), 292 mg/ml L-glutamine (Cat #SH30034.01; HyClone), 1 mM sodium pyruvate (Cat #11360-070; Gibco), 1X MEM nonessential amino acids (Cat #11140-050; Gibco), 25 mM Hepes, pH 7.3 (Cat #SH30237.01; HyClone), and 50 $\mu$M 2-mercaptoehthanol (Cat #M3180; Sigma-Aldrich).

The splenocytes were subjected to red blood cell lysis using ammonium chloride-potassium lysis buffer (150 mM $NH_4Cl$, 10 mM $KHCO_3$, 0.1 mM EDTA) for 4 min at room temperature. Thereafter, the lysis was quenched by adding 5 ml cRPMI and pelleting down the cells at 300$g$, 5 min. Pellet was then resuspended using 5 ml cRPMI. Cells were counted using a Beckman Coulter Z1 particle counter.

## Sample preparation for flow cytometry

0.5–2 × $10^6$ cells per sample were used. Live/Dead staining, cell–surface staining, and intracellular staining were performed using U-bottomed 96-well plates, with 100 $\mu$l per sample of antibody dilution used for staining, and 200 $\mu$l per sample of buffer used for washes.

For live/dead staining, cells were first washed once with 1X PBS and then stained with LIVE/DEAD fixable near-IR dead cell stain (Cat #L10119; Invitrogen) at a 1:1,000 (in 1X PBS) dilution of the reconstituted dye according to the recommended protocol. 100 $\mu$l of the live/dead stain was used per sample and incubated for 15 min on ice in the dark. After that, the cells were washed once with 1X PBS and then further surface staining was performed.

For cell surface staining, a cocktail of the desired antibodies was prepared with each of them diluted at a concentration of 1:300 in FACS wash buffer (FWB, 0.5% bovine serum albumin in PBS). 100 $\mu$l/sample of this antibody solution was added to the cells and incubated for 30 min in the dark on ice. After that, the samples were washed using 200 $\mu$l/sample FWB and were ready to further carry out intracellular staining or to be analyzed on flow cytometry.

For intracellular staining, after the cells were stained for cell–surface antigens, they were fixed with fixation buffer (Cat #88-8824-00; eBioscience) for 20 min at room temperature after which they were washed twice with 1X permeabilization buffer (Cat #88-8824-00; eBioscience). The permeabilized cells were then stained with Themis antibody at a dilution of 1:100 or cytokines (IFN$\gamma$ and TNF) at a dilution of 1:300 in the permeabilization buffer for 1 h or 30 min on ice, respectively, followed by a wash with 1X permeabilization buffer. The cells were then resuspended in FWB and analyzed on flow cytometer.

For intracellular staining of nuclear antigens, an eBioscience Foxp3/Transcription Factor Staining Buffer Set (Cat #00-5523-00) was used. After staining for cell-surface antigens, each sample was fixed/permeabilized in 500 $\mu$l Foxp3 fixation/permeabilization working solution for 1 h on ice, followed by a wash in 1X permeabilization solution, and incubation with antibody against the nuclear antigen in 100 $\mu$l of 1X permeabilization solution on ice for 1 h. The samples were then washed once with 1X permeabilization solution, followed by a final wash and re-suspension in FWB.

For metabolic maker analysis, 0.5–1 × $10^6$ T cells were spun down at 300$g$ for 5 min at 4°C. Cells were first fixed with 1 ml of 4% PFA (Cat #50-980-487; Electron Microscopy Sciences) which was added simultaneously along with vortexing followed by incubating them at room temperature for 15 min. Then, the cells were washed thoroughly two times with 2 ml of FWB. Thereafter, the cells were permeabilized by adding 1 ml of ice-cold 90% methanol while vortexing the cells, followed by 30 min incubation on ice. After two washes, the samples were incubated in the dark for 1 h at room temperature with 50 $\mu$l of 1:100 (c-Myc and p-Akt) and 1:200 (p-S6)

diluted primary antibodies. After one wash, the samples were in 50 $\mu$l of FWB containing fluorescent conjugated secondary antibody at a dilution of 1:2,000 and antibodies against cell surface markers at 1:300 dilution for 30 min at room temperature in dark. Cells were washed and then analyzed on a flow cytometer.

Flow cytometry analysis was performed on Fortessa X-20 (BD), with FACSDiva used for acquisition and FlowJo versions 9 and 10 used for analysis.

## Flow antibody list

The following antibodies from BD, eBioscience and BioLegend were used from cell surface antigen staining: CD8$\alpha$ (clone 53-6.7; eBioscience and BD), CD5 (clone 53-7.3; eBioscience), CD25 (clone 7D4; BD), CD44 (clone IM7; eBioscience), CD69 (clone H1.2F3; BioLegend and eBioscience), CD62L (clone MEL-14; eBioscience and BD), CD98 (clone RL388; BioLegend), CD122 (clone 5H4; eBioscience), V$\alpha$2 (clone B20.1; eBioscience), CD107a (clone 1D4B; BD), and TCR$\beta$ (clone H57-597; BD).

For intracellular staining, the following antibodies from eBioscience were used: TNF (clone MP6-XT22), IFN$\gamma$ (clone XMG1.2), and Themis (clone 1TYMS).

For nuclear staining, the following antibodies were used: Foxp3 (clone MF-14; BioLegend), GATA3 (clone TWAJ, eBioscience), and Eomes (clone Dan11mag; eBioscience).

For metabolic marker analysis, the following primary antibodies from Cell Signaling were used: c-Myc (clone D3N8F), pAkt T308 (clone D25E6), and p-S6 ribosomal protein S235/236 (clone D57.2.2E). Goat anti-rabbit F(ab')2 fragment Alexa Fluor 488 (Invitrogen) was used as secondary antibody.

## CTV labeling

To assess the proliferative capacity of T cells, the sorted naïve cells were labeled with CTV (Life Technologies). For this, T cells were adjusted at a concentration of 2 × $10^6$/ml in PBS. Then, the cell suspension was stained with CTV dye at a concentration of 5 $\mu$M. Immediately, after the addition of CTV, the cells were vortexed. They were then incubated at 37°C for 10 min with vortexing cells after every 5 min. At the end of incubation, the media-containing serum at five times the original staining volume was added to the cells to quench the reaction. Cells were then pelleted at 300$g$, 4°C for 5 min and cell count was done manually to check the live cell numbers to set up the experiment.

## T cell activation assays

For OT-I Themis cWT and cKO TX: All T cell activation assays were performed in cRPMI. 1 × $10^5$ sorted naïve (CD44$^{lo}$) OT-I CD8$^+$ were added per well in U-bottomed 96-well plates and incubated with the OVA APLs (0.5 $\mu$g/ml) for 3 h or 24 h at 37°C, 5% $CO_2$, followed by cell surface antibody staining for flow analysis. For proliferation, CTV labeled 1 × $10^5$ sorted naïve cells were stimulated with OVA APLs (0.1 $\mu$g/ml) for 2–3 d. After this, the cell supernatant was collected for performing IL-2 ELISA and the samples were stained with the desired markers for analysis on flow cytometer.

For metabolic assays, $1 \times 10^6$ cells per well were plated in 24-well plate and stimulated with OVA APLs (0.5 µg/ml) for 24 h at 37°C, 5% $CO_2$. After that, the cells were assessed for c-Myc, p-Akt, p-S6, and Seahorse assays were performed.

For CTL culture, OT-I splenocytes were resuspended in 40 ml of cRPMI containing OVA peptide (10 ng/ml; NIH) and recombinant human IL-2 (rhIL-2) (10 U/ml; [Cat #202-IL; R&D Systems]) at 37°C for 24 h in a vertically placed T-75 flask. The T cells were then pelleted the next day and cultured further with only rhIL-2 containing media for 2 d to induce their proliferation and to lower their activation state post antigenic stimulation. CTL were used after 2 d.

### Cytokine production assay

CTL were stimulated with OVA APLs presented by EL4 as antigen-presenting cells, at a ratio of 3 ($3.5 \times 10^6$/ml):1 ($1.5 \times 10^6$/ml) (T cells: EL4) for 6 h at 37°C in the presence of Golgiplug (Brefeldin A) (Cat #555029; BD) used at a 1:1,000 dilution. After 6 h, they were assessed for IFNγ- and TNF-positive cells using flow cytometry.

### Cytotoxicity assay

For this assay, EL4 were used as target cells. They were loaded with 1 µg/ml OVA APLs in cRPMI for 1 h at 37°C and washed two times using at least twice the original volume in cRPMI to remove unloaded peptides. After that, the EL4 cells loaded with OVA and its APLs were labeled with 5 µM CTV, whereas EL4 cells loaded with non-stimulatory peptide VSV were labeled with 0.5 µM of CTV. After that, OT-I CTL and peptide-loaded EL4 cells were co-cultured in cRPMI at an effector-to-target ratio of 10 ($3.5 \times 10^6$/ml): 1 ($0.355 \times 10^6$/ml) in a U-bottomed 96-well plate to a final volume of 200 µl per well. EL4 cells loaded with VSV were added in the same ratio as the other peptide-loaded EL4 cells and were used as internal controls in every sample. Cells were incubated for 4 h at 37°C before live/dead staining and analysis on flow cytometry. Percent specific lysis was calculated according to the formula:

$$\% \, specific \, lysis = \left( 1 - \frac{no. \, of \, live \, EL4 \, (peptides)}{no. \, of \, live \, EL4 \, (VSV)} \right) \times 100$$

### Preparation of cell lysates

$2 \times 10^6$ cells CTL were taken and pelleted down. The cells were resuspended in 500 µl of appropriate media in an eppendorf tube and then either left unstimulated or stimulated with 5 µl OVA APLs tetramers for 2 min. Post stimulation, cold PBS was added, and cells were pelleted. The supernatant was removed, and cells were lysed with 50 µl of lysis buffer (pH 7.5) (1% maltoside [Cat #D4641; Sigma-Aldrich], 20 mM TrisHCl, 150 mM NaCl, pH 7.5 with Halt Protease & Phosphatase Inhibitor Cocktail [Cat #1861280; Thermo Fisher Scientific]) for about 45 min placed in a cold orbital shaker set at 4°C. After that, the cells were spun down at 13,000g for 15 min at 4°C. The cell debris and rest of insoluble components formed a pellet, whereas the cell lysate was obtained from the supernatant, which was transferred. Reducing sample buffer (Cat #09499-14; Nacalai Tesque) was added to the lysates, to attain 1X concentration and

the lysates were boiled for 3 min at 96°C. ERK and Shp1 phosphorylation were analyzed using western blotting.

### Western blot

10 µl of the whole-cell lysate was loaded into a pre-cast 4–12% gradient SDS gel (Cat #NP0323BOX; Invitrogen) and 1X MOPS buffer was used to run the gel. After this, the proteins were transferred onto an Immobilon-FL membrane (Cat #IPFL00010; Millipore) which was methanol activated for about 3 min. The protein transfer was carried out at 100 V for 1 h. After transfer, membrane was blocked for 1 h at room temperature in LI-COR blocking buffer (Cat #927-40100; Li-Cor). The membrane was thereafter probed for the target proteins using respective primary antibodies as per recommended dilutions prepared in the blocking buffer and was incubated overnight in a cold room on the orbital shaker. Then, the blots were incubated for 1 h at room temperature in dark on the orbital rocker with a mixture of secondary antibodies. LI-COR Odyssey infrared imaging system was used for detection.

Primary antibodies are as follows: Anti-Themis rabbit polyclonal (Cat #06-1328; Merck Millipore), GAPDH (clone 14C10, Cat #2118; Cell Signaling Technology), pERK (p-p44/p-42 MAPK Erk1 Y204/Erk2 Y187, clone D1H6G, Cat #5726S; Cell Signaling Technology), and total Erk1 (clone MK12, Cat #610031; BD). Secondary antibodies are as follows: goat anti-mouse IgG IRDye800 (Cat #926-32210; Li-Cor) and goat anti-rabbit IgG IRDye680 (Cat #926-68021; Li-Cor).

### In vitro 4-hydroxy tamoxifen (4-OH tamoxifen) treatment

Splenocytes harvested from OT-I mice were stimulated with 10 nM OVA peptide (SIINFEKL) in cRPMI for 24 h. Stimulated T cells were treated with 4 µM of (Z)-4-OHT (Cat #H7904; Sigma-Aldrich), and were cultured for two more days at a cell concentration of $0.2–0.5 \times 10^6$ cells/ml in cRPMI supplemented with 10 U/ml rhIL-2 (Cat#202-IL; R&D Systems) in T75 flasks.

### OCR and ECAR drug concentrations

OCR and ECAR were measured in XF media (non-buffered RPMI 1640 containing 25 mM glucose [only for OCR], 2 mM L-glutamine, and 1 mM sodium pyruvate) under basal conditions and in response to various drugs. For OCR, the levels were measured in response to 1 µM oligomycin, 1.5 µM fluoro-carbonyl cyanide phenylhydrazone (FCCP), and 1 µM rotenone + 1 µM antimycin A. For ECAR, the levels were measured in response to 10 mM glucose, 1 µM oligomycin, and 50 mM 2-deoxy-D-glucose.

### Mitochondrial analysis

T cells ($0.5–1 \times 10^6$ cells) were centrifuged at 300g for 5 min at 4°C and washed with PBS. Then, the cells were incubated at 37°C for 35–40 min in the dark with 125 µl of 50 nM of MitoTracker Green FM (Cat #M7514; Invitrogen) diluted in pre-warmed media without serum. After that, two washes were given with PBS to remove the remaining Mitotracker and analysis was done on flow cytometer.

## 3D T cell functional assay in a microfluidic device

B16-OVA and B16-Q4R7 transfected to express nuclear GFP (Plasmid #58468; Addgene) were counted and resuspended at $2.5 \times 10^6$ cells/ml in a 2.5 mg/ml collagen type I (Cat #354236; Corning) solution at pH 7.4. Collagen solution was prepared and inserted in microfluidic devices (AIM Biotech) following a previously described protocol (Pavesi et al, 2017). The microfluidic device consists into three parallel microchannels. The central channel is delimited by triangular posts to retain the hydrogel solution by surface tension. The cell-loaded hydrogel solution containing either the target cells B16-OVA or the B16-Q4R7 was injected into the central gel channel (Fig 6A) and allowed to polymerize for 25 min in a 5% $CO_2$ incubator at 37°C. After polymerization, fluidic channels were hydrated with cRPMI supplemented with 20 U/ml rhIL-2 (Cat #202-IL; R&D Systems). OT-I Themis cWT and Themis cKO TX CTL were labeled with CTV (Cat #C34557; Thermo Fisher Scientific) at 5 $\mu$M working concentration. After labeling, CTL were counted, resuspended at $8 \times 10^6$ cells/ml, and inserted in one of the lateral fluidic channels. After 24 h in a 5% $CO_2$ incubator at 37°C, cells in the device were stained with a live/dead fixable staining (Cat #PK-CA707-32007; Promokine) following the vendor protocol and then incubated at room temperature for 15 min with Image-iT Fixative Solution containing 4% formaldehyde and methanol-free (Cat #FB002; Life Technologies). After washing steps, devices were filled with PBS and imaged with a FV3000RS confocal microscope (Olympus).

## Statistical analysis

Unpaired $t$ test, one-way ANOVA or two-way ANOVA tests were used for comparison between two genotypes, calculated by Prism8 (GraphPad). The sample size and definition of n are indicated in each figure legend. All the datasets follow normal distribution.

# Data Availability

The data are available from the corresponding author upon request.

# Supplementary Information

# Acknowledgements

This research was supported by the Singapore Ministry of Education's grant MOE-000112, the Singapore Ministry of Health's National Medical Research Council under its CBRG/0097/2015 and the VICTORY OF-LCG. We would like to thank Prof. Toshio Suda (Cancer Science Institute, NUS, Singapore) for the Cre-ERT2 strain, and Doctors Paul Hutchinson and Teo Huo Gui from NUS Immunology Programme, Flow Cytometry Core facility for cell sorting.

## Author Contributions

N Gautam: investigation, visualization, methodology, and writing—original draft, review, and editing.
L Wojciech: investigation, visualization, methodology, and writing—original draft, review, and editing.
J Yap: investigation.
YL Chua: investigation.
EMW Ding: investigation.
DCN Sim: investigation.
ASM Tan: investigation and methodology.
PJ Ahl: investigation and methodology.
M Prasad: methodology.
DWH Tung: investigation.
JE Connolly: resources, supervision, and funding acquisition.
G Adriani: resources, supervision, visualization, and writing—original draft.
J Brzostek: conceptualization, supervision, and writing—review and editing.
NRJ Gascoigne: conceptualization, supervision, funding acquisition, and writing—original draft, review, and editing.

## Conflict of Interest Statement

The authors declare that they have no conflict of interest.

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
