## [Reviewer comments · Life Science Alliance]

Life Science Alliance

Themis controls T cell activation, effector functions, and metabolism of peripheral CD8⁺ T cells

Namrata Gautam, Lukasz Wojciech, Jiawei Yap, Yen Chua, Eyan Ding, Don Sim, Alrina Tan, Patricia Ahl, Mukul Prasad, Desmond Tung, John Connolly, Giulia Adriani, Joanna Brzostek, and Nicholas Gascoigne

DOI: <https://doi.org/10.26508/lsa.202302156>

Corresponding author(s): Nicholas Gascoigne, National University of Singapore

Review Timeline:

Submission Date:	2023-05-14
Editorial Decision:	2023-07-10
Revision Received:	2023-08-20
Editorial Decision:	2023-09-07
Revision Received:	2023-09-14
Accepted:	2023-09-14

Transaction Report:

July 10, 2023

Re: Life Science Alliance manuscript #LSA-2023-02156-T

Prof. Nicholas R.J. Gascoigne
National University of Singapore
Department of Microbiology
5 Science Drive 2
Singapore 117597

Dear Dr. Gascoigne,

Thank you for submitting your manuscript entitled "Themis controls T cell activation, effector functions, and metabolism of peripheral CD8+ T cells" to Life Science Alliance. The manuscript was assessed by expert reviewers, whose comments are appended to this letter. We invite you to submit a revised manuscript addressing the Reviewer comments.

Thank you for this interesting contribution to Life Science Alliance. We are looking forward to receiving your revised manuscript.

Sincerely,

B. MANUSCRIPT ORGANIZATION AND FORMATTING:

Reviewer #1 (Comments to the Authors (Required)):

Themis deletion has an important impact on CD8 T cell maturation in the thymus but it was found not to affect T cell activation of mature peripheral T cells. Here, Gascoigne and colleagues provide an explanation for this lack of effect of Themis deficiency and that is of adaptation of T cells to long-term deficiency by upregulating or downregulating a considerable number of regulators of T cell activation downstream the TCR. They have created an inducible system for the downregulation of Themis and show that soon after downregulation T cells show widespread defects in T cell activation through the TCR. An important finding is that Themis is necessary for the regulation of T cell metabolites which is required for T cell proliferation and clonal expansion. Overall I think it is an interesting work that clarifies a long-lasting conundrum. I think that the work has been correctly carried out and I do not have any major concern about its quality for publication.

Reviewer #2 (Comments to the Authors (Required)):

The main question addressed by this paper by Gaotam et al is the role of Themis in peripheral CD8 T cells using a mouse Cre-ERT2 or distal Ick mediated conditional knockout of the gene. The authors use in vitro stimulation of OT-I CD8 T cells with OVA derived altered peptides or loaded tetramers (to accomplish varying ligand affinities) to compare the WT or Themis deficient CD8 T cells in various aspects (TCR proximal signaling events, in vitro proliferation, tumor cell killing in a microfluidic tumor model, transcriptomes). The authors state that "proliferation, effector functions including anti-tumor killing, and upregulation of energy metabolism are all severely compromised". The authors find that this in contrast with the effects of long-term deletion of Themis (in the current and in a previous study) and suggest that mice can adapt to long-term loss of Themis. The paper also address the question of how Themis regulates SHP-1 - question addressed in other studies with contradicting conclusions. The authors conclude that Themis acts as a negative regulator of SHP-1. The group has published previously in this area including a recent paper this year (Tang et al 2023) using an LCMV infection of Themis conditional knockout mice. The experiments have been designed well, performed in replicates and statistical analysis has been provided where needed. The data support the statements made by the authors although for the in vitro proliferation experiments, its a bit tough to evaluate the biological relevance of the difference seen (20% for Q4R7). No additional experiments are suggested.

My comments are listed below.

Major comments.

1. This is a well written paper, but for a wider reach to immunologists who work in different areas, I suggest the paper be improved for clarity. Currently, it is a bit confusing to read and the specific question addressed by the authors (that has not been addressed by them or others previously) is not obvious. The findings, if possibly summarized in a schematic could make easier reading.
2. Can the authors comment on the unique aspects of the in vitro stimulated OT-I T cells to address their question as opposed to the in vivo LCMV infection (Tang et al 2023) that they have used previously?
3. given that several studies (as stated by the authors) have addressed the effector of Themis on SHP-1, can the authors comment on why this particular study has the definitive answer or the conditions under which differing effects on SHP-1 may occur?

Minor comments

1. In figure 3, I suggest showing the flow cytometry data and gating strategy.
2. In figure 3, can the authors provide a reason for the two different times (4 or 6 hours) use dfor stimulation?
3. In figure 2, was the CD25 expression tested after 3 days too? Please specify time point if not.
4. In figure A (western blot) is it possible to provide a summary of the data from multiple experiments instead of a representative plot for rigor?

Point-by-point

Reviewer #1 (Comments to the Authors (Required)):

Themis deletion has an important impact on CD8 T cell maturation in the thymus but it was found not to affect T cell activation of mature peripheral T cells. Here, Gascoigne and colleagues provide an explanation for this lack of effect of Themis deficiency and that is of adaptation of T cells to long-term deficiency by upregulating or downregulating a considerable number of regulators of T cell activation downstream the TCR. They have created an inducible system for the downregulation of Themis and show that soon after downregulation T cells show widespread defects in T cell activation through the TCR. An important finding is that Themis is necessary for the regulation of T cell metabolids which is required for T cell proliferation and clonal expansion. Overall I think it is an interesting work that clarifies a long-lasting conundrum. I think that the work has been correctly carried out and I do not have any major concern about its quality for publication.

Response:

We thank the reviewer for these positive comments.

Reviewer #2 (Comments to the Authors (Required)):

The main question addressed by this paper by Gaotam et al is the role of Themis in peripheral CD8 T cells using a mouse Cre-ERT2 or distal Ick mediated conditional knockout of the gene. The authors use in vitro stimulation of OT-I CD8 T cells with OVA derived altered peptides or loaded tetramers (to accomplish varying ligand affinities) to compare the WT or Themis deficient CD8 T cells in various aspects (TCR proximal signaling events, in vitro proliferation, tumor cell killing in a microfluidic tumor model, transcriptomes). The authors state that "proliferation, effector functions including anti-tumor killing, and upregulation of energy metabolism are all severely compromised". The authors find that this in contrast with the effects of long-term deletion of Themis (in the current and in a previous study) and suggest that mice can adapt to long-term loss of Themis. The paper also address the question of how Themis regulates SHP-1 - question addressed in other studies with contradicting conclusions. The authors conclude that Themis acts as a negative regulator of SHP-1. The group has published previously in this area including a recent paper this year (Tang et al 2023) using an LCMV infection of Themis conditional knockout mice. The experiments have been designed well, performed in replicates and statistical analysis has been provided where needed. The data support the statements made by the authors although for the in vitro proliferation experiments, its a bit tough to evaluate the biological relevance of the difference seen (20% for Q4R7). No additional experiments are suggested.

Response:

We thank the reviewer for these positive comments. Regarding the "tough to evaluate" part regarding the relatively small reduction in divided cells with Q4R7 (Fig. 2b), we think this is just one of the many cases where loss of Themis seems to affect relatively weak ligands (e.g Q4H7) more than it does stronger ligands (e.g. Q4R7). This is mentioned in the text.

My comments are listed below.

Major comments.

1. This is a well written paper, but for a wider reach to immunologists who work in different areas, I suggest the paper be improved for clarity. Currently, it is a bit confusing to read and the specific question addressed by the authors (that has not been addressed by them or others previously) is not obvious. The findings, if possibly summarized in a schematic could make easier reading.

Response:

We have now made a schematic as a “graphical abstract”. We have also added a few sentences in abstract, significance statement, intro, and discussion to clarify.

2. Can the authors comment on the unique aspects of the in vitro stimulated OT-I T cells to address their question as opposed to the in vivo LCMV infection (Tang et al 2023) that they have used previously?

Response:

Please note that the Tang paper also used the chronic (dLck-Cre) Themis deletion model, so the acute deletion model that is the subject of this paper is different and novel. We added this sentence to the Discussion: “In the dLck-Cre-mediated cKO model in OT-I mice, we found negligible reduction in TCR stimulation with OVA APLs but altered memory versus effector cell differentiation in responses to acute viral infection (Brzostek et al., 2020; Tang et al., 2023).”

3. given that several studies (as stated by the authors) have addressed the effector of Themis on SHP-1, can the authors comment on why this particular study has the definitive answer or the conditions under which differing effects on SHP-1 may occur?

Response:

We think that the Themis story is still revealing its complexities. We have added the following text to the Discussion: “Hence, our data indicate that Themis acts as a negative regulator of Shp1 in peripheral T cells, as proposed by Love and colleagues for thymocytes (Choi et al., 2017a; Choi et al., 2017b; Zvezdova et al., 2016). However, it is becoming clear that Themis acts not only through Shp1 regulation of TCR signals, but also through effects on PD-1 (Kinosada et al., 2017; Tang et al., 2023) and CD137 signaling, as well as through cytokine responsiveness (Liu et al., 2022). These separate but integrated responses are not necessarily through a single biochemical mechanism.”

Minor comments

1. In figure 3, I suggest showing the flow cytometry data and gating strategy.

Response:

We have included flow cytometry data and gating strategy for the data shown in Figure 3 as new Supplementary Figure 4. Additionally, we have provided a detailed explanation in the figure legend for a better understanding of the assays performed.

2. In figure 3, can the authors provide a reason for the two different times (4 or 6 hours) used for stimulation?

Response:

This was because of the different requirements of the cytotoxicity versus the intracellular cytokine staining assays. We used the 4 hour time point to analyze cytotoxicity of the OT-I CTL as the cytotoxicity can be easily detected then by staining the cells with the LIVE/DEAD Fixable cell stain (as mentioned in the supplementary methods section). However, for cytokine production and CD107a production, analyzed by flow cytometry, we used the 6 hour time point. As mentioned in the supplementary methods section, we used brefeldin A (BD, Cat# 555029) to block intracellular protein transport, resulting in the accumulation of cytokines in the Golgi complex. This intracellular accumulation of cytokines enhances the detection of cytokine-producing cells. The accumulation of cytokines requires time, depending upon the type of cells. Hence, we chose 6 hours to easily detect the cytokine positive population and CD107a expression to correlate the two markers. Moreover, we used different affinity OVA APLs. Therefore, using a late time point made sure that the cytokine production could be detected even after stimulation with the lower affinity peptide. This gave us the best opportunity to capture a snapshot of the cytokine and CD107a expression upon stimulation with different OVA APLs between the two Themis genotypes. Additionally, according to the

manufacturers' protocol, brefeldin A addition can even be optimally used up to 12 hours to detect the cytokine-producing population, emphasising that our experimental method is well-accepted in the immunology field.

3. In figure 2, was the CD25 expression tested after 3 days too? Please specify time point if not.

Response:

Yes, the CD25 expression was tested after 3 days of stimulation with different OVA APLs. The information is updated in the main text as well as in Figure 2 legend, for both CD25 expression and IL-2 production.

4. In figure A (western blot) is it possible to provide a summary of the data from multiple experiments instead of a representative plot for rigor?

Response:

We have included a summary of the reduction in phosphorylation of Shp1 and ERK1/2 in the OT-I Themis cKO TX sample as compared to the OT-I Themis cWT sample in the form of bar graphs. This is presented as Supplementary Figure 2.

September 7, 2023

RE: Life Science Alliance Manuscript #LSA-2023-02156-TR

Prof. Nicholas R.J. Gascoigne
National University of Singapore
Department of Microbiology
5 Science Drive 2
Singapore 117597

Dear Dr. Gascoigne,

Thank you for submitting your revised manuscript entitled "Themis controls T cell activation, effector functions, and metabolism of peripheral CD8+ T cells". We would be happy to publish your paper in Life Science Alliance pending final revisions necessary to meet our formatting guidelines.

- Please add a Category for your manuscript in our system
- Please add the Twitter handle of your host institute/organization as well as your own or/and one of the authors in our system
- Please use the [10 author names, et al.] format in your references (i.e., limit the author names to the first 10)
- We encourage you to revise the figure legends for figures 1, 6, S2, and S3 such that the figure panels are introduced in alphabetical order
- please add callouts for Figures 5G,H; 6I,J; S1A-C; S4A-C; S5A-B to your main manuscript text
- the Supplementary Material should be incorporated into the main Materials and Methods section

A. FINAL FILES:

B. MANUSCRIPT ORGANIZATION AND FORMATTING:

Sincerely,

Reviewer #2 (Comments to the Authors (Required)):

In this revised version, the authors have addressed all my comments satisfactorily. The graphical abstract was helpful and it is not referenced in the paper but hopefully this or a simpler version can be included in the final version.

Point-by-point

Reviewer #1 (Comments to the Authors (Required)):

Themis deletion has an important impact on CD8 T cell maturation in the thymus but it was found not to affect T cell activation of mature peripheral T cells. Here, Gascoigne and colleagues provide an explanation for this lack of effect of Themis deficiency and that is of adaptation of T cells to long-term deficiency by upregulating or downregulating a considerable number of regulators of T cell activation downstream the TCR. They have created an inducible system for the downregulation of Themis and show that soon after downregulation T cells show widespread defects in T cell activation through the TCR. An important finding is that Themis is necessary for the regulation of T cell metabolidsms which is required for T cell proliferation and clonal expansion. Overall I think it is an interesting work that clarifies a long-lasting conundrum. I think that the work has been correctly carried out and I do not have any major concern about its quality for publication.

Response:

We thank the reviewer for these positive comments.

Reviewer #2 (Comments to the Authors (Required)):

The main question addressed by this paper by Gaotam et al is the role of Themis in peripheral CD8 T cells using a mouse Cre-ERT2 or distal Ick mediated conditional knockout of the gene. The authors use in vitro stimulation of OT-I CD8 T cells with OVA derived altered peptides or loaded tetramers (to accomplish varying ligand affinities) to compare the WT or Themis deficient CD8 T cells in various aspects (TCR proximal signaling events, in vitro proliferation, tumor cell killing in a microfluidic tumor model, transcriptomes). The authors state that "proliferation, effector functions including anti-tumor killing, and upregulation of energy metabolism are all severely compromised". The authors find that this in contrast with the effects of long-term deletion of Themis (in the current and in a previous study) and suggest that mice can adapt to long-term loss of Themis. The paper also address the question of how Themis regulates SHP-1 - question addressed in other studies with contradicting conclusions. The authors conclude that Themis acts as a negative regulator of SHP-1. The group has published previously in this area including a recent paper this year (Tang et al 2023) using an LCMV infection of Themis conditional knockout mice. The experiments have been designed well, performed in replicates and statistical analysis has been provided where needed. The data support the statements made by the authors although for the in vitro proliferation experiments, its a bit tough to evaluate the biological relevance of the difference seen (20% for Q4R7). No additional experiments are suggested.

Response:

We thank the reviewer for these positive comments. Regarding the "tough to evaluate" part regarding the relatively small reduction in divided cells with Q4R7 (Fig. 2b), we think this is just one of the many cases where loss of Themis seems to affect relatively weak ligands (e.g Q4H7) more than it does stronger ligands (e.g. Q4R7). This is mentioned in the text.

My comments are listed below.

Major comments.

1. This is a well written paper, but for a wider reach to immunologists who work in different areas, I suggest the paper be improved for clarity. Currently, it is a bit confusing to read and the specific question addressed by the authors (that has not been addressed by them or others previously) is not obvious. The findings, if possibly summarized in a schematic could make easier reading.

Response:

We have now made a schematic as a “graphical abstract”. We have also added a few sentences in abstract, significance statement, intro, and discussion to clarify.

2. Can the authors comment on the unique aspects of the in vitro stimulated OT-I T cells to address their question as opposed to the in vivo LCMV infection (Tang et al 2023) that they have used previously?

Response:

Please note that the Tang paper also used the chronic (dLck-Cre) Themis deletion model, so the acute deletion model that is the subject of this paper is different and novel. We added this sentence to the Discussion: “In the dLck-Cre-mediated cKO model in OT-I mice, we found negligible reduction in TCR stimulation with OVA APLs but altered memory versus effector cell differentiation in responses to acute viral infection (Brzostek et al., 2020; Tang et al., 2023).”

3. given that several studies (as stated by the authors) have addressed the effector of Themis on SHP-1, can the authors comment on why this particular study has the definitive answer or the conditions under which differing effects on SHP-1 may occur?

Response:

We think that the Themis story is still revealing its complexities. We have added the following text to the Discussion: “Hence, our data indicate that Themis acts as a negative regulator of Shp1 in peripheral T cells, as proposed by Love and colleagues for thymocytes (Choi et al., 2017a; Choi et al., 2017b; Zvezdova et al., 2016). However, it is becoming clear that Themis acts not only through Shp1 regulation of TCR signals, but also through effects on PD-1 (Kinosada et al., 2017; Tang et al., 2023) and CD137 signaling, as well as through cytokine responsiveness (Liu et al., 2022). These separate but integrated responses are not necessarily through a single biochemical mechanism.”

Minor comments

1. In figure 3, I suggest showing the flow cytometry data and gating strategy.

Response:

We have included flow cytometry data and gating strategy for the data shown in Figure 3 as new Supplementary Figure 4. Additionally, we have provided a detailed explanation in the figure legend for a better understanding of the assays performed.

2. In figure 3, can the authors provide a reason for the two different times (4 or 6 hours) used for stimulation?

Response:

This was because of the different requirements of the cytotoxicity versus the intracellular cytokine staining assays. We used the 4 hour time point to analyze cytotoxicity of the OT-I CTL as the cytotoxicity can be easily detected then by staining the cells with the LIVE/DEAD Fixable cell stain (as mentioned in the supplementary methods section). However, for cytokine production and CD107a production, analyzed by flow cytometry, we used the 6 hour time point. As mentioned in the supplementary methods section, we used brefeldin A (BD, Cat# 555029) to block intracellular protein transport, resulting in the accumulation of cytokines in the Golgi complex. This intracellular accumulation of cytokines enhances the detection of cytokine-producing cells. The accumulation of cytokines requires time, depending upon the type of cells. Hence, we chose 6 hours to easily detect the cytokine positive population and CD107a expression to correlate the two markers. Moreover, we used different affinity OVA APLs. Therefore, using a late time point made sure that the cytokine production could be detected even after stimulation with the lower affinity peptide. This gave us the best opportunity to capture a snapshot of the cytokine and CD107a expression upon stimulation with different OVA APLs between the two Themis genotypes. Additionally, according to the

manufacturers' protocol, brefeldin A addition can even be optimally used up to 12 hours to detect the cytokine-producing population, emphasising that our experimental method is well-accepted in the immunology field.

3. In figure 2, was the CD25 expression tested after 3 days too? Please specify time point if not.

Response:

Yes, the CD25 expression was tested after 3 days of stimulation with different OVA APLs. The information is updated in the main text as well as in Figure 2 legend, for both CD25 expression and IL-2 production.

4. In figure A (western blot) is it possible to provide a summary of the data from multiple experiments instead of a representative plot for rigor?

Response:

We have included a summary of the reduction in phosphorylation of Shp1 and ERK1/2 in the OT-I Themis cKO TX sample as compared to the OT-I Themis cWT sample in the form of bar graphs. This is presented as Supplementary Figure 2.

September 14, 2023

RE: Life Science Alliance Manuscript #LSA-2023-02156-TRR

Prof. Nicholas R.J. Gascoigne
National University of Singapore
Department of Microbiology
5 Science Drive 2
Singapore 117597

Dear Dr. Gascoigne,

Thank you for submitting your Research Article entitled "Themis controls T cell activation, effector functions, and metabolism of peripheral CD8+ T cells". It is a pleasure to let you know that your manuscript is now accepted for publication in Life Science Alliance. Congratulations on this interesting work.

DISTRIBUTION OF MATERIALS:

Again, congratulations on a very nice paper. I hope you found the review process to be constructive and are pleased with how the manuscript was handled editorially. We look forward to future exciting submissions from your lab.

Sincerely,
